# Inhibiting endocytosis in CGRP$^+$ nociceptors attenuates inflammatory pain-like behavior

Rasheen Powell [1], Violet A. Young[2], Kerri D. Pryce [1], Garrett D. Sheehan[2], Kwaku Bonsu[1], Abdulelah Ahmed [1] & Arin Bhattacharjee [1,2 ✉]

The advantage of locally applied anesthetics is that they are not associated with the many adverse effects, including addiction liability, of systemically administered analgesics. This therapeutic approach has two inherent pitfalls: specificity and a short duration of action. Here, we identified nociceptor endocytosis as a promising target for local, specific, and long-lasting treatment of inflammatory pain. We observed preferential expression of AP2α2, an α-subunit isoform of the AP2 complex, within CGRP$^+$/IB4$^-$ nociceptors in rodents and in CGRP$^+$ dorsal root ganglion neurons from a human donor. We utilized genetic and pharmacological approaches to inhibit nociceptor endocytosis demonstrating its role in the development and maintenance of acute and chronic inflammatory pain. One-time injection of an AP2 inhibitor peptide significantly reduced acute and chronic pain-like behaviors and provided prolonged analgesia. We evidenced sexually dimorphic recovery responses to this pharmacological approach highlighting the importance of sex differences in pain development and response to analgesics.

[1] Pharmacology and Toxicology, University at Buffalo - The State University of New York, Buffalo, NY 14203, USA. [2] Program for Neuroscience, University at Buffalo - The State University of New York, Buffalo, NY 14203, USA. ✉email: ab68@buffalo.edu

The physiology of inflammatory pain involves the integration of primary afferent neurons, the central nervous system, and the immune system[1–5]. Peripheral sensitization of dorsal root ganglion (DRG) nociceptors initiates inflammatory pain and is driven by inflammatory mediators released from immune cells and damaged tissue[1,4,6]. Recently, calcitonin gene-related peptide (CGRP) containing nociceptors were identified as principal coordinators of thermal and mechanical sensitivity in various pain models[7,8]. Therefore, it is reasonable to consider CGRP$^+$ nociceptors as potential analgesic targets.

There is an unmet need for efficacious analgesics with lesser adverse effects. Opioid drugs, the most widely prescribed class of medications in the United States, are commonly used for pain treatment. In addition to their high potential for addiction, there are concerns that opioids can lead to hypotension, sleep apnea, reduced hormone production and, in the elderly, increased falls leading to hip fractures. Opioids also cause respiratory depression, and there is now an ever-increasing concern over the intersection of the opioid epidemic with the Covid-19 pandemic[9,10]. Other treatment options for inflammatory pain include non-steroidal anti-inflammatory drugs and corticosteroids, but they have been increasingly contraindicated for extended use due to detrimental side effects[11,12]. Nociceptive ion channel inhibitors seemed to be attractive molecules for analgesia; however, they have demonstrated limited clinical efficacy and are not currently used as a treatment option[13]. After screening more than three-thousand transgenic mouse knockout lines, the endocytosis associated-adaptin protein kinase 1 (AAK1), was considered a putative target for pain treatment sparking small molecule development to inhibit this enzyme[14]. Targeting AAK1 systemically, however, might be problematic due to its ubiquitous expression and further development of AAK1 inhibitors for pain relief has yet to be pursued. Nonetheless, this study[14] did mark the first pre-clinical attempt to provide analgesia by pharmacologically inhibiting endocytosis.

The primary endocytic machinery in neurons utilizes the multimeric adaptor protein complex 2 (AP2), which exhibits differential expression of its α-subunit isoforms in neurons: the α1 isoform localizes to synaptic compartments, whereas the α2 isoform displays considerable extra-synaptic expression[15]. Previously, we have shown that AP2 clathrin-mediated endocytosis (AP2-CME) underlies DRG neuronal sensitization through internalization of sodium-activated potassium channels ($K_{Na}$) in vitro and that the AP2α2 subunit becomes associated with these channels after protein kinase A (PKA) stimulation[16]. In this system, pharmacological inhibition of endocytosis using a lipidated AP2 inhibitor prevented PKA-induced DRG neuronal hyperexcitability. These findings were the first to implicate endocytosis as necessary for DRG neuronal sensitization. Whether extra-synaptic endocytosis in DRG neurons is required for pain initiation and pain maintenance have yet to be addressed.

In the present study, we locally disrupt nociceptor endocytosis and use various inflammatory pain models to characterize the in vivo contribution of extra-synaptic AP2-CME to inflammatory pain. We provide further evidence for peptidergic nociceptors as principal regulators of inflammatory pain. Our study highlights the ability of lipidated peptidomimetics to target superficial nerve afferents and to provide long-lasting analgesia. Additionally, we describe sexually dimorphic differences in pain-like behavior during inflammation across pain models and animal species.

## Results

### AP2α2 is preferentially expressed in CGRP containing DRG neurons.
Previous immunological labeling of AP2α2 in the superficial lamina of the rodent dorsal horn suggested a putative differential expression of AP2α2 in nociceptors[3,17–19]. In order to resolve this, we probed mDRG neurons with antibodies against AP2α2[20], CGRP, and an Alexa fluor-conjugated IB4. Interestingly, we observed strong immunofluorescent co-localization between CGRP and AP2α2 while virtually no IB4$^+$ neurons expressed AP2α2 (Fig. 1a). The overlapping immunoreactivity of AP2α2 and CGRP, alongside a lack of immunoreactivity in IB4$^+$ neurons, suggested AP2α2 participates in peptidergic DRG neuronal signaling implicating it in thermal sensitivity during pain[8].

### In vivo DRG neuronal AP2α2 knockdown modulates peripheral nociceptor excitability and reduces acute inflammatory pain-like behaviors.
CGRP expression is a strong marker for thermal nociceptors due to robust co-expression of the transient receptor potential vanilloid 1 (TRPV1) ion channel[21–23]. TRPV1 is known to principally govern nociceptor responses to noxious thermal and chemical sensation as well as acidic pH[24,25]. Inflammation-induced ongoing pain is therefore driven by TRPV1 nociceptive fibers[26]. Observing a high degree of co-expression of AP2α2 and CGRP suggested that AP2α2 contributes to thermal and chemical responsiveness. To test this, a unilateral injection of shRNAs against AP2α2 was made into the sciatic nerve of C57BL/6 mice. This produced a significant decrease in AP2α2 protein expression levels 7 days post shRNA injection (Fig. 1b, c) and was sufficient in reducing PKA-induced hyperexcitability in dissociated adult DRG neurons from these mice (Fig. 1d). Dissociated contralateral IB4$^-$ DRG neurons demonstrated firing accommodation under control conditions ($n = 10$; only 2 of 10 exhibited more than 2 action potentials, Fig. 1d $top$). Ipsilateral IB4$^-$ DRG neurons cultured from scrambled shRNA animals displayed typical loss of firing accommodation under PKA stimulatory conditions ($n = 7$; 5/7 hyperexcitable, Fig. 1d $middle$). Ipsilateral DRG neurons cultured from AP2α2 shRNA animals displayed firing accommodation ($n = 9$; 2/9 hyperexcitable Fig. 1d $bottom$).

The behavioral consequence of in vivo DRG neuronal AP2α2 knockdown was first assessed using the formalin acute inflammatory pain assay. The biphasic nature of this assay offers compartmentalization of observed behavioral effects to distinct neurophysiological changes[27–29]. DRG neuronal knockdown of AP2α2 did not alter transient phase 1 pain-like behaviors (Fig. 1e); however, there was a significant decrease in inflammatory phase 2-paw licking (scrambled shRNA 406 ± 70; AP2 shRNA 193 ± 73) and lifting behaviors (Fig. 1e; scrambled shRNA 246 ± 23; AP2 shRNA 99 ± 43). Additionally, there were time-dependent changes in animal resting behavior (Fig. 1f). At the start of observation, in phase 1, both groups exhibit increased paw licking behaviors, indicative of pain (Fig. 1f $left$). However, at the start of phase 2, the scrambled shRNA group continued to engage in paw licking, whereas the AP2α2 shRNA group engaged in grooming behavior (Fig. 1f $middle$). Finally, at the conclusion of observation, the scrambled group had maintained paw licking behavior, while the AP2α2 group began to exhibit exploratory behavior (Fig. 1f $right$; red arrow). Representative videos of this behavior are provided for reference in the Supplemental Files.

To evaluate the contribution of AP2α2 to chronic inflammatory pain, we conducted an intraplantar injection of Complete Freund's Adjuvant (CFA). CFA induces hypersensitivity and local inflammation through immune cell recruitment and activation. Using this model, we were able to evaluate the contribution of endocytosis in the development (AP2 knockdown pre-inflammation Fig. 2a) and maintenance (AP2 knockdown post-inflammation Fig. 2c) of chronic inflammatory nociceptor signaling. In the pre-inflammation condition, knockdown of AP2α2 occurred 7-days before CFA injection. A pre-

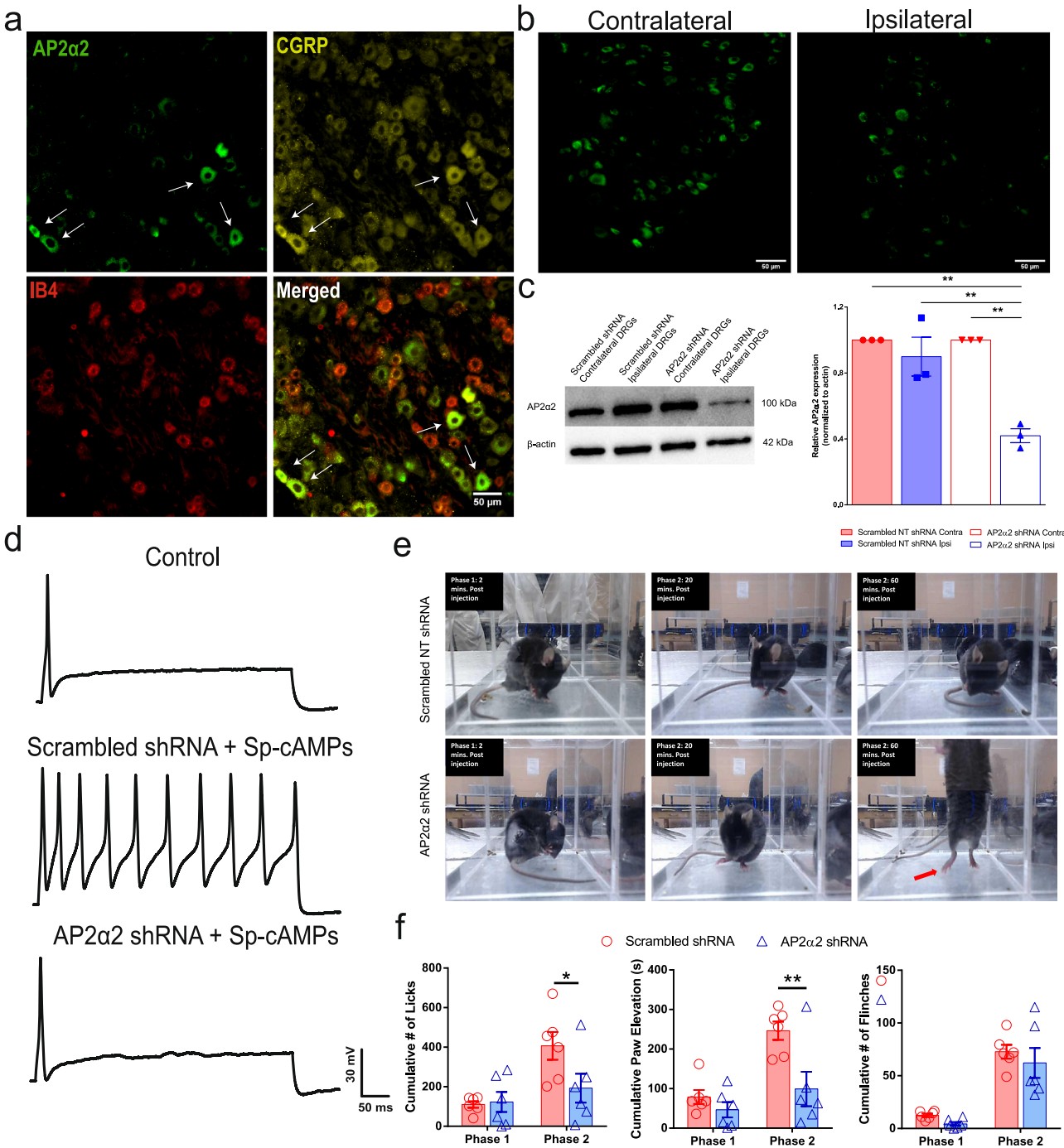

inflammation knockdown of AP2α2 ($n = 12$) produced a decrease in thermal sensitivity following CFA injection (4.2 ± 0.8 s), whereas control animals ($n = 11$) exhibited sensitivity to thermal stimuli 24 h following CFA injection (2.3 ± 0.3 s). The difference in paw withdrawal latencies persisted for the duration of experimentation until both groups displayed full recovery of thermal sensitivity. We then conducted shRNA-mediated AP2α2 knockdown experiments following inflammation, where genetic knockdown of AP2α2 occurred 4 days following CFA injection (Fig. 2c). AP2α2 knockdown animals recovered more rapidly ($n = 8$, day-5: 7.0 ± 0.6 s; day-9: 7.5 ± 0.6 s; day-13: 8.1 ± 0.8 s) compared to control shRNA animals ($n = 8$, day-5: 6.2 ± 0.7 s; day-9: 5.2 ± 0.5 s; day-13: 5.4 ± 0.8 s). Due to the localization of AP2α2 in peptidergic neurons, we did not expect knockdown to alter von Frey thresholds, but to our surprise, we observed a slight

reduction in von Frey thresholds when AP2α2 was preemptively knocked down, significantly at day 13 (Fig. 2b). Contralateral von Frey responsiveness data is shown in Supplementary Fig. 2. These data suggested that DRG neuronal AP2α2 knockdown disrupted neuroplastic processes necessary for both thermal and mechanical sensitivity during inflammation.

**Lipidated peptidomimetics localize to lipid compartments in the rodent hind paw.** We[30] and others[31] have previously utilized small myristoylated peptides to target nociceptor endings and modify pain-like behavior. Small lipidated peptides are able to traverse the membrane by a flip-flop mechanism gaining access to the inside of the cell (Supplementary Fig. 1). Here we wanted to explore how lipidated peptides might enter nociceptive nerve endings and persist in cells and tissues after administration as we

**Fig. 1 AP2α2 is expressed in IB4-, CGRP+ nociceptors, and in vivo AP2α2 knockdown attenuates nocifensive behaviors. a** Representative immunofluorescent image showing expression patterns of AP2α2 and CGRP. IB4 reactivity was used to delineate between peptidergic and non-peptidergic DRG neurons. AP2α2 is preferentially expressed in small- and medium-sized CGRP+ DRG neurons but not in IB4+ neurons. Arrows highlight strong co-localization of CGRP and AP2α2. This experiment was repeated independently 3 times and similar results were obtained. **b** AP2α2 immunoreactivity in the ipsilateral DRG after in vivo AP2α2 knockdown compared to contralateral DRG taken from the same animal, seven days after knockdown. This experiment was repeated independently 3 times, and similar results were obtained. **c** [Left] Representative Western blot showing extent of AP2α2 knockdown. Paired contralateral and ipsilateral samples are taken from the same animal. [Right] Densitometry analysis of Western blots. Samples were collected from biologically independent animals ($n = 3$ for scrambled shRNA and $n = 3$ for AP2α2 shRNA). Animals were sacrificed seven days after knockdown. Data is presented as mean pixel density ± S.E.M. Significance determined by one-way ANOVA with Holms-Sidak correction $p < 0.05$; *$p < 0.01$; **. Scrambled shRNA Contra vs AP2α2 shRNA Ipsi $p$-value = 0.0011. Scrambled shRNA Ipsi vs AP2α2 Ipsi shRNA $p$-value = 0.0025. AP2α2 shRNA Contra vs AP2α2 shRNA Ipsi $p$-value = 0.0011. **d** Representative traces from dissociated adult DRG neurons recorded under varying conditions: [Top] Control conditions, [Middle] DRG neurons transfected with Scrambled shRNAs following 30-min exposure to Adenosine-3′, 5′- cyclic monophosphate, Sp- isomer (Sp-cAMPs), [Bottom] DRG neurons transfected with AP2α2 shRNAs during PKA stimulatory conditions. IB4- were selectively recorded as determined by absence of fluorescent after incubation with an IB4-alexa fluor 488 conjugate. **e** Representative images depicting pain-like behaviors in C57BL/6 mice 2 min [left], 20 min [middle], and 60 min [right] post-formalin injection. Red arrow highlights use of inflamed ipsilateral paw. **f** Summarized nocifensive behaviors from C57BL/6 mice following injection with 5% formalin. Phase 1; 0–10 min, phase 2; 11–60 min post injection (scrambled shRNA group $n = 6$; AP2α2 group $n = 6$). Data is pooled males and females and is presented as cumulative means ± S.E.M. Significance determined using repeated measures 2-way ANOVA with Bonferroni correction $p < 0.05$; *$p < 0.01$; **. Cumulative # of licks: Phase 2 Scrambled shRNA vs AP2α2 shRNA $p$-value = 0.0313. Cumulative Paw Elevation: Phase 2 Scrambled shRNA vs AP2α2 shRNA $p$-value = 0.0026.

will later describe the use of a lipidated AP2 inhibitor peptide in our studies. We generated a lipidated version of the influenza hemagglutinin (HA) protein (HA-peptide) and visualized its localization by immunocytochemistry. We found that the HA-peptide embedded into the membranes of CHO cells resulting in robust membrane labeling (Supplementary Fig. 3). This was striking considering the conditions of the experiment; exposure to HA-peptide for 3 h followed by a series of washes and media replacement. The persistence of HA-immunoreactivity over time (at least 72 h) was equally surprising, which suggested that small lipidated peptides maintain a degree of stability during large cellular events such as mitosis. The persistence of lipidated peptides was similarly observed in cultured DRG neurons, detected 72 h after initial application and a series of media changes (Supplementary Fig. 3).

Next, we determined whether the lipidated HA-peptide could similarly demonstrate stability when applied in vivo, and whether inflammation impacts absorption and distribution of the peptide. Injection of the HA-peptide into the hind paw of mice produced robust HA immunoreactivity within the dermis and lipid dense compartments, while the epidermis and muscle displayed weak immunoreactivity 24 h after local injection (Fig. 3a). We specifically noted the presence of HA-immunoreactivity in nerve-like fibers in the dermis (Fig. 3a-1) as well as the muscle tissue (Fig. 3a-2). The presence of the HA-peptide in muscle localized nerve-like fibers suggested that lipidated peptides are able to laterally diffuse along the length of the fiber. A similar pattern of distribution was also observed under inflammatory conditions (Fig. 3b). There was considerable labeling of nerve-like fibers innervating the dermis (Fig. 3b-1) and muscle (Fig. 3b-2) under non-inflammatory conditions. Under inflammatory conditions we noted more intense global immunoreactivity (Fig. 3b-4). The cytoplasmic pan-neuronal marker PGP9.5 was used to confirm the presence of the HA-peptide in peripheral neurons (Fig. 3c).

**AP2 inhibitor peptide attenuated pain-like behaviors during inflammation.** The consequences of pharmacologically inhibiting endocytosis in peripheral nociceptor afferents during inflammation was assessed using a small lipidated AP2-CME inhibitor peptide[32]. We unilaterally injected a short peptide derived from the human CD4 di-leucine motif with a myristoyl moiety conjugated to the N-terminal (Supplementary Table 1) 24 h before administering the formalin assay. This peptide sequence was shown to have high affinity (650 nM) for the AP2 complex[33].

One-time injection of the lipidated AP2 inhibitor peptide produced a robust decrease in cumulative phase 2 paw licking behavior (scrambled peptide $n = 6$, 184 ± 22; AP2 inhibitor peptide $n = 6$, 89 ± 23) while other measures of pain-like behavior remained relatively unchanged (Fig. 4a). Representative videos of this behavior are provided for reference. From this, we were able to recapitulate some of the reduced pain-like behaviors observed in the AP2α2 knockdown experiments, reinforcing the premise that local nociceptor endocytosis is participating in the development of inflammatory pain.

We then studied the analgesic potential of the AP2 inhibitor peptide during established CFA-induced inflammatory pain. First, we induced CFA inflammation for 24 h and then delivered a simple one dose injection of peptide directly into the inflamed paw. This single injection of the AP2 inhibitor peptide ($n = 16$) produced a persistent increase in paw withdrawal latency that persisted over a 4-day period (day-2: 5.4 ± 0.5 s; day-3: 7.5 ± 0.5 s; day-5: 7.9 ± 0.6 s) whereas the scrambled peptide group ($n = 16$, day-2: 3.3 ± 0.4 s; day-3: 4.5 ± 0.4 s; day-5: 5.9 ± 0.5 s) exhibited a prototypical thermal responsiveness recovery curve expected in a CFA model of inflammatory pain[34–36] (Fig. 4b). Interestingly, after segregating the data by gender, we noted an unexpected sex-dependent temporal component underlying the onset of analgesia (Fig. 4e). Comparison of the male ($n = 8$) and female ($n = 8$) AP2 inhibitor groups revealed that male mice responded more rapidly to the AP2 inhibitor peptide (Fig. 4e; 24 h) compared to females but this observation was not sustained for the duration of the experiment (Fig. 4e; 96 h). Male mice displayed an immediate and significant increase in paw withdraw latency 24 h following AP2 inhibitor peptide injection when compared to male mice that received the scrambled peptide (Supplementary Fig. 4C) whereas female mice that received the AP2 inhibitor peptide displayed paw withdraw latencies indistinguishable from the scrambled peptide group 24 h post-peptide injection (Supplementary Fig. 4H). Area under the curve (A.U.C.) quantification of the recovery curve (days 1–13) revealed that the AP2 inhibitor peptide produced an analgesic-like effect (Fig. 4c) in both male (Supplementary Fig. 4D) and female (Supplementary Fig. 4I) mice. If measured by absolute magnitude of effect, male mice responded more favorably to the AP2 inhibitor peptide when compared to female mice (male A.U.C.$_{AP2}$ = 101.4 ± 3.3 v.s. female A.U.C.$_{AP2}$ = 82.2 ± 4.5). In order to gain a deeper understanding of the sex differences between male and female mice during thermal recovery, we chose to express the rate of

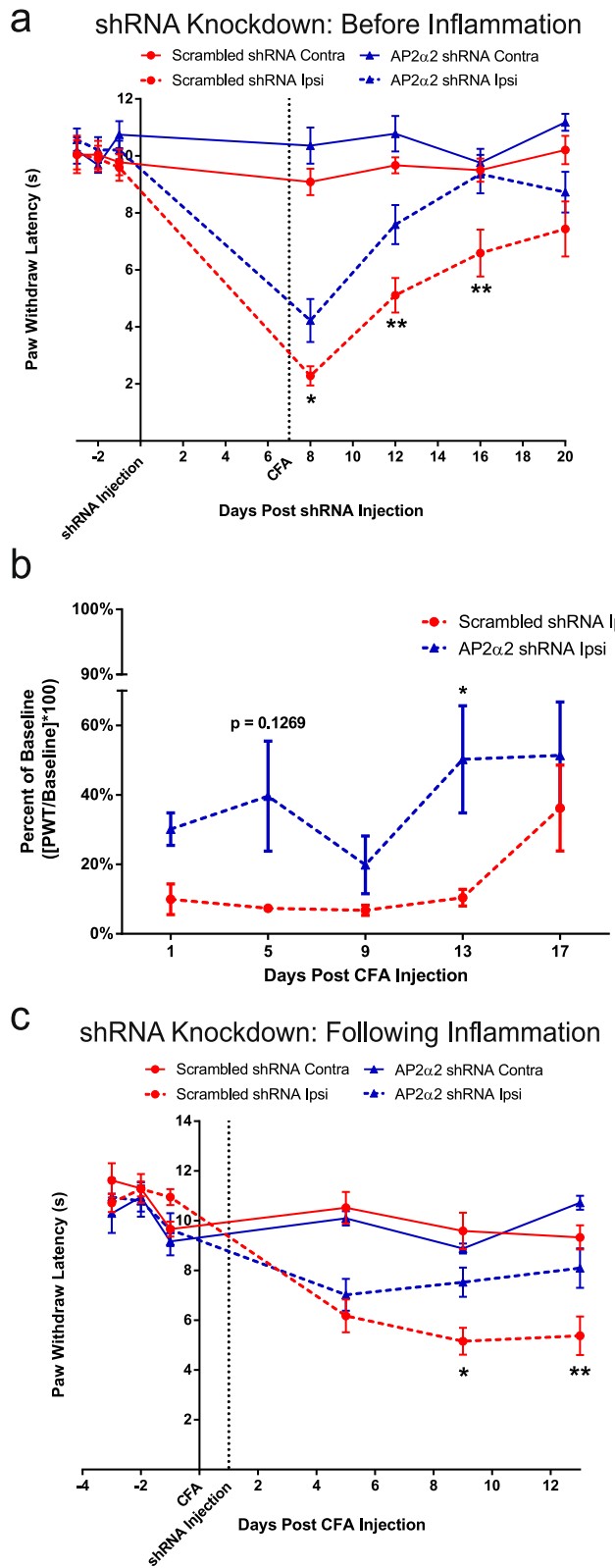

**Fig. 2 AP2α2 knockdown disrupts the development and maintenance of thermal sensitivity in chronic inflammatory pain. a** Thermal sensitivity of animals in the CFA pain model with a pre-inflammatory knockdown paradigm for scrambled ($n = 11$) and AP2α2 ($n = 12$) shRNA groups. Data is pooled males and females and is represented as mean paw withdrawal latency (PWL) ± S.E.M. Significance determined using repeated measures 2-way ANOVA with Bonferroni correction $p < 0.05$; *$p < 0.01$; ** Day 8: Ipsilateral Scrambled shRNA vs Ipsilateral AP2α2 shRNA $p$-value = 0.0398. Day 12: Ipsilateral scrambled shRNA vs Ipsilateral AP2α2 shRNA $p$-value = 0.0048. Day 16: Ipsilateral scrambled shRNA vs ipsilateral AP2α2 shRNA $p$-value = 0.0013. **b** von Frey withdrawal threshold of ipsilateral hind paw in a chronic inflammatory pain model where knockdown occurred before inflammation was initiated. Data for the scrambled ($n = 8$) and AP2α2 ($n = 8$) shRNA groups is pooled, males and females, and is represented as mean percentage of baseline ± S.E.M. Significance determined using repeated measures 2-way ANOVA with Bonferroni correction $p < 0.05$; *. Day 13: Scrambled shRNA vs AP2α2 shRNA $p$-value = 0.0315. Contralateral PWT can be found in Supplementary Fig. 2. **c** Thermal sensitivity of animals injected with scrambled ($n = 8$) and AP2α2 ($n = 8$) shRNAs following establishment of inflammation. Data is pooled males and females and is represented as mean PWL ± S.E.M. Significance determined using repeated measures 2-way ANOVA with Bonferroni correction $p < 0.05$; *$p < 0.01$; **. Day 9: Ipsilateral scrambled shRNA vs ipsilateral AP2α2 shRNA $p$-value = 0.0251. Day 13: Ipsilateral scrambled shRNA vs Ipsilateral AP2α2 shRNA $p$-value = 0.0064.

unprecedented; however, our implementation of a one-phase exponential decay approach allowed for a more comprehensive understanding of the kinetics of thermal recovery following a single-dose administration paradigm, which may prove important for the development of clinically relevant analgesics. Taken together, application of the AP2 inhibitor peptide resulted in a more rapid thermal sensitivity recovery ($\tau_{AP2} = 2.12$) compared to control ($\tau_{control} = 4.29$; Fig. 4d). Again, separation of the composite data set by sex uncovered a difference between male and female recovery kinetics during inflammation. Male mice experienced a strong decrease in tau (Supplementary Fig. 4E; $\tau_{control} = 8.24$, $\tau_{AP2} = 1.93$), while female mice maintained similar tau values between scrambled and AP2 peptide groups (Supplementary Fig. 4J; $\tau_{control} = 2.92$, $\tau_{AP2} = 2.45$). Administration of the AP2 inhibitor peptide slightly affected von Frey withdraw thresholds (Fig. 4g). Using a dynamic weight bearing apparatus, we were able to measure weight borne on each paw following injury which functions as a measure of non-evoked inflammatory hyperalgesia[37-40]. In this assay, weight borne on the inflamed hind paw was significantly increased 48 h following AP2 peptide injection (Fig. 4f). We chose to look at 48 h, because of the delayed response to AP2 inhibition in females (Fig. 4e and Supplementary Fig. 4H). Altogether these data show that pharmacological inhibition of endocytosis directly impacted thermal responsiveness and had consequential effects on weight bearing and indirect effects on mechanical sensitivity.

In addition to chemogenic-induced inflammation, we also explored the analgesic potential of the AP2 inhibitor peptide in injury-induced inflammation using a post-operative pain model in rats[41]. Preclinical incision models are useful for determining the efficacy of pharmacologic treatment during the early postsurgical phase[42]. For this assay, we simulated a potential clinical application schedule for the AP2 inhibitor peptide; large-volume (200 µL) sub-cutaneous administration into the ipsilateral hind paw 6 h before incision, followed by a series of small-volume sub-cutaneous (25 µL) and intra-muscular (50 µL) injections immediately following incision (Fig. 5a). These small volumes

recovery as a time constant. Since thermal sensitivity exhibits time-dependent recovery in CFA models of pain[34-36], we used the scrambled peptide group recovery phase (day 1–day 11) as a measure of unassisted resolution of thermal sensitivity. By fitting this curve to a first-order exponential decay equation, we calculated a time constant, tau ($\tau$). Determination of analgesia in animal models through quantification of recovery is not

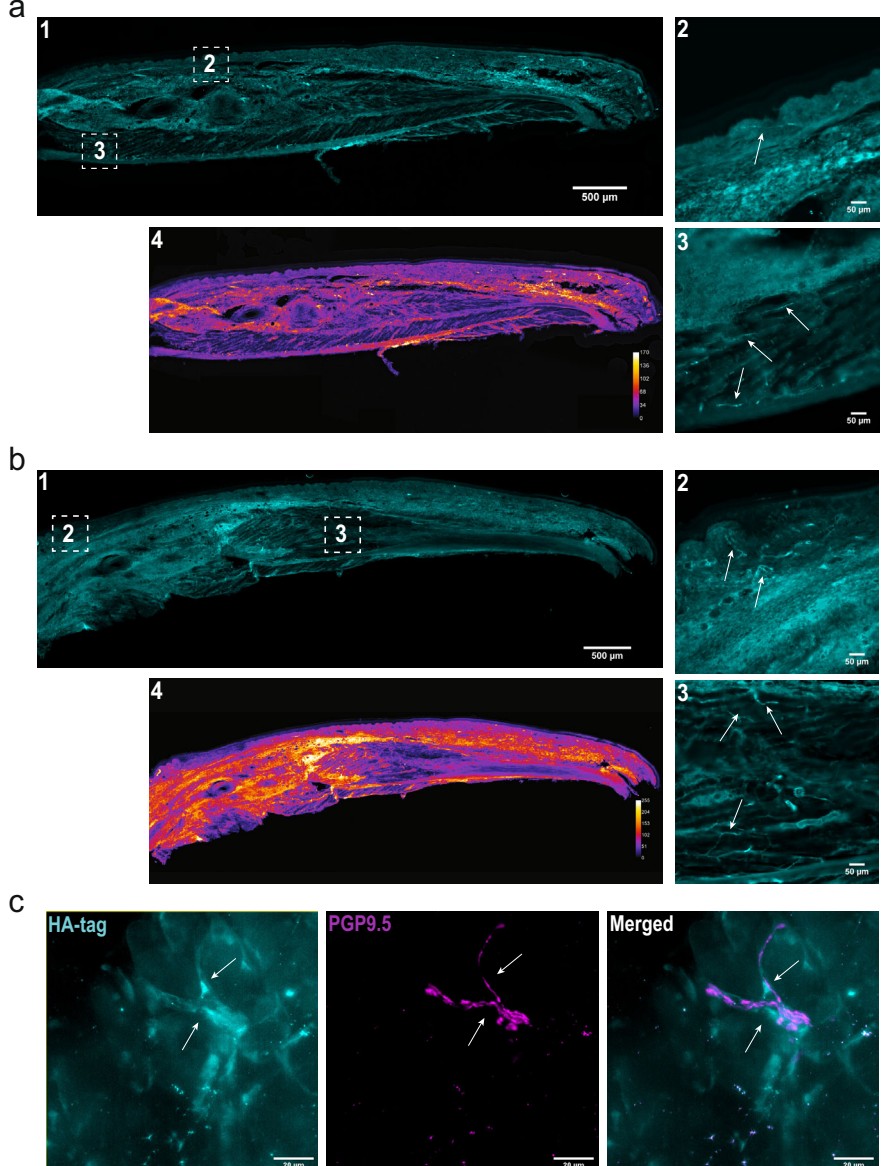

**Fig. 3 Lipidated peptides infiltrate peripheral neuronal afferents. a** Tilescan immunofluorescent image of a mouse hind paw following injection of an antigenic lipidated-HA peptide into the hind paw of a naïve C57BL/6 mouse under control conditions. The HA-peptide allowed for immunofluorescent visualization of lipidated peptide distribution following injection. This experiment was repeated, independently, twice with similar results. **a4** Heatmap analysis of immunoreactivity. **a1** The lipidated-HA peptide preferentially partitioned to the dermis, within lipid dense areas. **a2** Insert depicting HA immunoreactivity in peripheral afferents within the dermis. **a3** Insert depicting immunoreactivity in peripheral afferents in muscle tissue. While afferents exhibited immunolabeling, muscle cells did not. **b** Injection of an antigenic lipidated peptidomimetic into the hind paw of a naïve C57BL/6 mouse under CFA-induced inflammation. This experiment was repeated, independently, twice with similar results. **b4** Heatmap analysis depicted greater retention of peptide within inflamed tissues. **b1** Again, lipidated-HA peptide preferentially partitioned to the dermis, specifically, lipid dense areas. **b2** Insert depicting immunoreactivity in peripheral afferents in the dermis. **b3** Insert depicting immunoreactivity in peripheral afferents and in muscle tissue. **c** Immunofluorescent image depicting localization of the membrane bound HA-peptide and the cytoplasmic pan-neuronal marker PGP9.5. This experiment was repeated, independently, twice with similar results. [Left] HA-peptide alone. Arrows highlight the contour of the fiber. [Middle] PGP9.5 alone. Arrows illustrate the contour of the neuron afferent observed with the HA-peptide. [Right] Merged image. Arrows show that the HA-peptide and PGP9.5 immunoreactivity follow similar contours.

were used to reduce leakage out of the wound. Application of the AP2 inhibitor peptide ($n = 18$) produced a profound and long-lasting reduction in thermal sensitivity compared to the scrambled peptide ($n = 14$; Fig. 5b). Just as in previous models, the AP2 inhibitor peptide was capable of significantly increasing thermal sensitivity thresholds, compared to the scrambled peptide, over a 5-day period following a single application (scrambled peptide day-1: 5.8 ± 0.4 s; day-2: 7.6 ± 0.7; day-3:

7.6 ± 0.5 s; day-4: 7.8 ± 0.5 s; day-5: 8.4 ± 0.4 s; day-6: 9.5 ± 0.6 s vs AP2 inhibitor peptide day-1: 8.2 ± 0.4 s; day-2: 9.3 ± 0.4; day-3: 10.1 ± 0.5 s; day-4: 10.5 ± 0.6 s; day-5: 11.2 ± 0.5 s; day-6: 10.8 ± 0.5 s). Again, after segregating the data based on gender, we noted sex-dependent differences. Male rats injected with the AP2 inhibitor peptide displayed a significant reduction in thermal sensitivity 24 h following injection, but the effect did not reach significance at subsequent time points (Supplementary Fig. 5A).

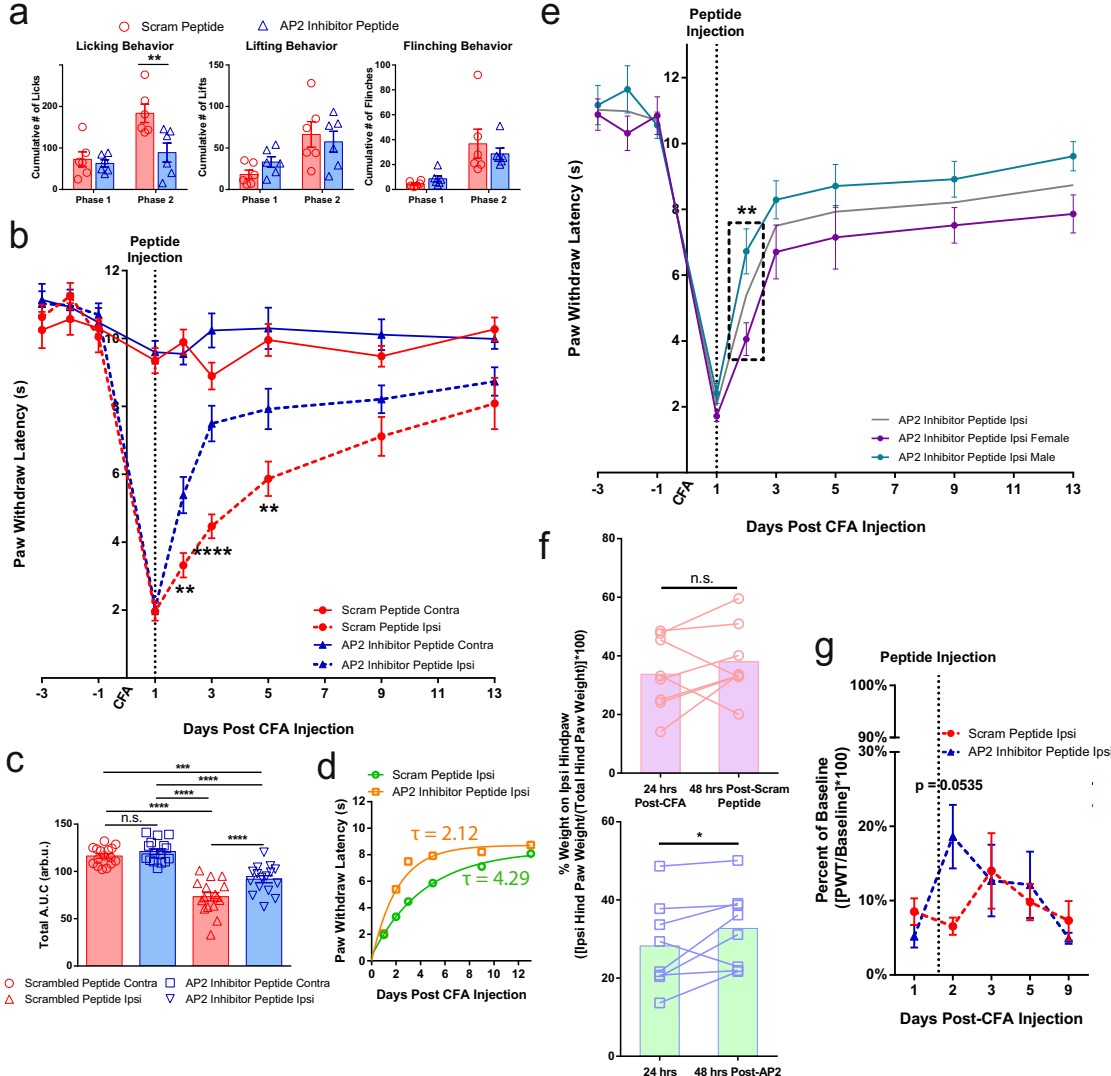

Remarkably, in females, the AP2 inhibitor peptide caused thermal responsiveness to return to baseline by as early as day-3 and persisted with significance over the course of the assay (Supplementary Fig. 5D). A.U.C. quantification revealed that the AP2 inhibitor peptide was significantly increased the A.U.C when compared to the scrambled peptide (Fig. 5c). Interestingly, this effect was sex-dependent: for males the AP2 inhibitor peptide trended but did not produce a statistically significant analgesic-like effect (Supplementary Fig. 5B), whereas females injected with the AP2 peptide exhibited a significant increase in A.U.C compared to females that received the scrambled peptide (Supplementary Fig. 5E). Additionally, the AP2 inhibitor peptide increased the rate of recovery following incision (Fig. 5d; $\tau_{control} = 10.53$, $\tau_{AP2} = 3.41$). In this parameter, both sexes showed an increased rate of recovery (Male: Supplementary Fig. 5C; $\tau_{control} = 8.12$, $\tau_{AP2} = 2.85$, Female: Supplementary Fig. 5F; $\tau_{control} = 12.73$, $\tau_{AP2} = 5.24$). In contrast to the CFA model, males and females displayed significant differences in their respective paw withdrawal latencies 24 h following injection; females in the scrambled peptide group displayed less paw withdrawal latencies 24 h following incision compared to males in the scrambled peptide group; however, this effect almost reverses 120 h following incision (Fig. 5e). In other words, the thermal recovery curve for females is flat in the scrambled group, whereas males show progressive recovery. This distinguishing feature in

males and females underscored the gender-difference response to the AP2 inhibitor peptide (Supplementary Fig. 5). Similar to the CFA model, there was a significant effect on thermal sensitivity, but there was no significant change in the mechanical withdraw threshold (Fig. 5f).

We further tested the efficacies of di-leucine-based peptides derived from other human proteins and saw sequence-dependent reductions in various nocifensive behaviors (Supplementary Fig. 6). However, afferents at the site of formalin injection receive the highest concentrations of formalin thus are more likely to undergo fixation, inactivation, and/or desensitization[43]. Therefore, the formalin assay inherently underestimates the analgesic potential of lipidated peptides designed to penetrate afferent endings. Moreover, genetic and pharmacological inhibition of endocytosis did not preclude edema (Supplementary Figs. 2 and 4) nor immune cell activation and infiltration (Supplementary Fig. 7).

**Intraplantar injection of the AP2 inhibitor peptide caused nociceptor CGRP retention within the superficial layers of the epidermis.** Peripheral nociceptor afferents were previously shown to terminate in structurally distinct tissue layers in the dermis and epidermis[44]. Specifically, CGRP+ nociceptor afferents were shown to terminate in the stratum spinosum (SS) layer. Localized inhibition of endocytosis for 24 h, under non-inflammatory

**Fig. 4 Pharmacological inhibition of peripheral endocytosis by a lipidated AP2 inhibitor peptide attenuated pain-like behaviors during acute and chronic inflammation. a** Summarized nocifensive behaviors from C57BL/6 mice following injection with 5% formalin. The AP2 inhibitor peptide inhibitor was locally injected into the hind paw 24 h before formalin injection. Phase 1; 0–10 min, phase 2; 15–60 min post-injection (Scrambled peptide group $n = 6$; AP2 inhibitor peptide group $n = 6$). Data is pooled male and female mice and is presented as mean ± S.E.M. Significance determined using repeated measures 2-way ANOVA with Bonferroni correction $p < 0.05$; *Cumulative # of Licks: Phase 2 Scrambled peptide vs AP2 inhibitor peptide $p$-value = 0.0040. **b** Thermal sensitivity of animals that received either the scrambled peptide ($n = 16$) or the AP2 inhibitor peptide ($n = 16$) during established CFA-induced inflammatory pain. Each group was injected with the respective peptide 24 h after CFA. Data is pooled male and female animals and is represented as mean PWL ± S.E.M. Significance determined using repeated measures 2-way ANOVA with Bonferroni correction $p < 0.01$; **$p < 0.005$; ***. Day 2: Ipsilateral scrambled peptide vs Ipsilateral AP2 inhibitor peptide $p$-value = 0.0027. Day 3: Ipsilateral scrambled peptide vs Ipsilateral AP2 inhibitor peptide $p$-value < 0.0001. Day 5: Ipsilateral scrambled peptide vs Ipsilateral AP2 inhibitor peptide $p$-value = 0.0057. **c** Recovery area under the curve (A.U.C.; bounded by days 1–13)) quantification for all animals under experimental conditions displayed in (B; Days 1–13); scrambled peptide ($n = 16$) and AP2 inhibitor peptide ($n = 16$). Data shown is pooled male and female animals and is represented as the mean A.U.C. (in arbitrary units; Arb.u.) ± S.E.M. Statistical significance was determined using one-way ANOVA with Holms-Sidak correction $p < 0.05$; *$p < 0.01$; **$p < 0.005$; ***$p < 0.001$; ****. Contralateral scrambled peptide vs Ipsilateral scrambled peptide $p$-value < 0.0001. Contralateral Scrambled peptide vs ipsilateral AP2 Inhibitor peptide $p$-value = 0.0006. Contralateral AP2 inhibitor peptide vs Ipsilateral Scrambled peptide $p < 0.0001$. Contralateral AP2 inhibitor peptide vs Ipsilateral AP2 inhibitor peptide $p < 0.0001$. Ipsilateral scrambled peptide vs ipsilateral AP2 inhibitor peptide $p$-value < 0.0001. Pharmacological inhibition of endocytosis generated a significant increase in A.U.C. for the ipsilateral paw. **d** Recovery curve (bounded by days 1–13) fit to an exponential decay equation. Fitting the recovery curves from **b** reveals that inhibition of endocytosis accelerated the rate of recovery indicated by a decrease in tau. **e** Graph depicting the thermal sensitivity of the ipsilateral paw in male ($n = 8$; green) and female ($n = 8$; purple) animals that received the AP2 inhibitor peptide. Gray line represents mean AP2 inhibitor peptide ipsilateral paw withdraw threshold from **b**. Boxed area corresponds to data point of interest. Data is represented as the mean paw withdrawal ± S.E.M. Significance was determined using a 3-way ANOVA with Fishers Least Significant Difference Post-hoc test $p < 0.5$; *$p < 0.01$; **. Male Ipsilateral AP2 peptide vs Female AP2 inhibitor peptide $p$-value = 0.0069. **f** Percent of hind paw weight borne on the ipsilateral paw 24 h following CFA injection and 48 h following peptide injection. [Top] Change in percent of hind paw weight borne on the ipsilateral hind paw of animals that received the scrambled peptide ($n = 8$). [Bottom] Change in percent of hind paw weight borne on the ipsilateral hind paw of animals that received the AP2 inhibitor peptide ($n = 8$). Injection of the AP2 inhibitor peptide was sufficient in increasing weight bearing efficiency of the ipsilateral hind paw compared to 24 h following CFA injection. Injection of the scrambled peptide did not accelerate weight bearing efficiency of the ipsilateral hind paw. Data is pooled male and female mice and is represented as mean weight borne on the ipsilateral hind paw. Significance determined using a one-tailed paired Student's $t$ test $p < 0.05$; *. 24 hours post-CFA vs 48 hours post-AP2 peptide $p$-value = 0.0483. **g** Ipsilateral von Frey withdraw threshold of animals in either the scrambled peptide ($n = 11$) or the AP2 inhibitor peptide ($n = 11$) groups following establishment of CFA-induced inflammatory pain. Data is pooled male and female mice and is represented as mean PWT (as a percentage of baseline PWT) ± S.E.M. Significance determined using repeated measures 2-way ANOVA with Bonferroni correction $p < 0.05$; * Contralateral PWT can be found in Supplementary Fig. 4.

conditions, resulted in visualization of CGRP immunoreactivity throughout the SS layer as well as immunoreactivity in the stratum granulosum (SG) ($n = 3$ mice), indicating decreased basal CGRP release (Fig. 6). These data suggest that CGRP nociceptor afferents actually extend far more superficially in the epidermis than previously thought[44]. However, we did not observe CGRP retention in peripheral fibers in animals that were administered the AP2 inhibitor peptide 24 h after the establishment of CFA-induced inflammation (Supplementary Fig. 8). Prior studies have shown that release of CGRP from primary afferent neurons is increased during the period of maximal hyperalgesia that accompanies peripheral inflammation[45] and therefore the AP2 inhibitor peptide given 24 h after CFA might not be expected to alter CGRP immunoreactivity in peripheral terminals. We also observed granuloma-like clustering of immune cells after incision injury (previously observed following CFA injection[34]) indicating possible alterations in immune cell coordination; however, the AP2 inhibitor peptide did not appear to interrupt the attraction of immune cells to the site of injury (Supplementary Fig. 8). The pathophysiological consequences of these granuloma-like artifacts in the pain models used are not currently known but also might be due to decreased CGRP release.

**Differential expression of AP2α2 in CGRP+ neurons was also observed in human DRG.** Human and mouse AP2α2 share ~98% amino acid identity, as evidenced by sequence alignment, suggesting a strong evolutionary pressure to preserve protein function. Here we conducted hDRG immunohistochemistry studies, from a single donor, probing for AP2α2 and CGRP and we observed that hDRG also exhibited AP2α2 differential expression within CGRP+ neurons (Fig. 6d). Therefore, human inflammatory pain likely also depends on AP2α2-mediated

nociceptor endocytosis and should lend itself to pharmacological manipulation by the lipidated AP2 inhibitor peptide. All AP2 targeted peptides in this study utilized sequences derived from human proteins (Supplementary Table 1).

## Discussion
Using a genetic and a pharmacological approach in non-transgenic animals, we have demonstrated that inhibition of extra-synaptic nociceptor endocytosis significantly alters inflammatory pain-like behaviors.

Our characterization of AP2α2 expression in mouse DRG neurons revealed that peptidergic IB4- neurons preferentially express AP2α2 (Fig. 1a). We also observed a high level of co-expression with CGRP in human DRG neurons (Fig. 6d), suggesting AP2α2 participates in CGRP+ nociceptor signaling. CGRP+ nociceptors package neuropeptides in large-dense core vesicles (LDCV). They are released in a $Ca^{2+}$-dependent manner[22,23,46], potentiating inflammation, nociception, and immune cell activation[46–48]. Also LDCVs can fully collapse upon fusion to the membrane[49]. A robust membrane retrieval mechanism, namely endocytosis, would be required after neuropeptide release to allow further LDCV release. The mechanism for AP2-CME in synaptic vesicular membrane retrieval is well-established[50–54] recycling membrane after synaptic vesicle release. The preferential expression of the extra-synaptic AP2α2 in IB4- neurons is likely due to a specific dependence of membrane retrieval after LDCV release that occurs outside of the synapse. Prominent CGRP immunoreactivity in the SG layer of the dermis following AP2 inhibitor peptide injection (Fig. 6) suggested that changes in pain-like behaviors can be partially attributed to disruptions in CGRP release mechanisms. Decoupling endocytosis from exocytosis, either through the genetic or pharmacological

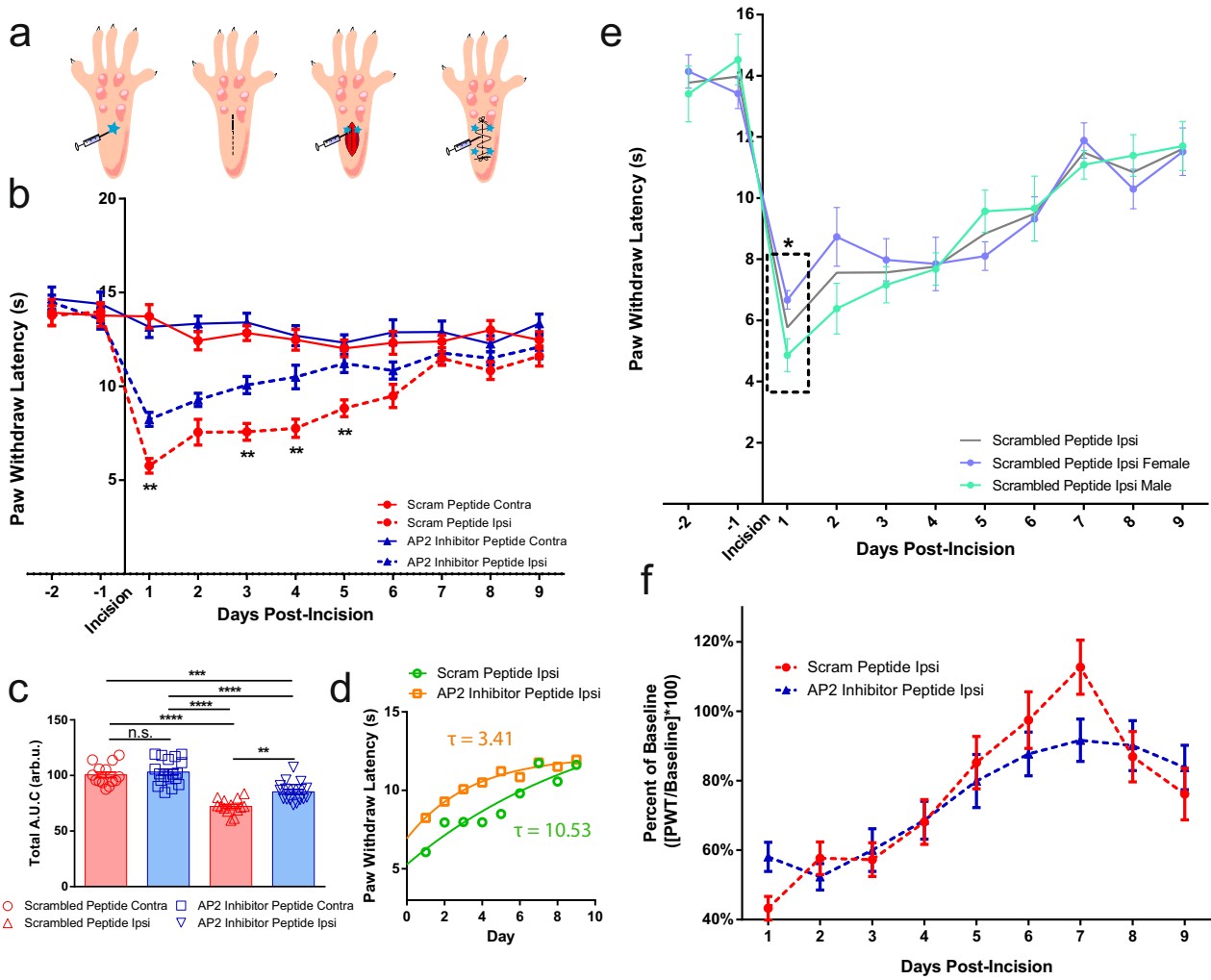

**Fig. 5 Pharmacological inhibition of peripheral endocytosis by a lipidated AP2 inhibitor peptide attenuated thermal sensitivity in a post-operative pain model. a** Schematic depicting injection protocol for lipidated AP2 inhibitor peptide. **b** Graph depicting thermal sensitivity in rats following plantar muscle incision and injection with scrambled peptide ($n = 14$) or AP2 inhibitor peptide ($n = 18$). Data shown is pooled males and females. Data is represented as mean PWL ± S.E.M. Significance determined using repeated measures 2-way ANOVA with Bonferroni correction $p < 0.05$; *$p < 0.01$; **. Ipsilateral scrambled peptide vs Ipsilateral AP2 peptide: Day 1 $p$-value = 0.0055, Day 3 $p$-value = 0.0048, Day 4 $p$-value = 0.0015, Day 5 $p$-value = 0.0082. **c** Total area under the curve (A.U.C.) quantification for all rats under experimental conditions displayed in **b**; scrambled peptide ($n = 14$) and AP2 inhibitor peptide ($n = 18$). Pharmacological inhibition of endocytosis generated a significant increase in A.U.C. for the ipsilateral paw. Data shown is pooled males and females. Data is represented as the mean A.U.C. (in arbitrary units; Arb.u.) ± S.E.M. Statistical significance was determined using one-way ANOVA with Holms-Sidak correction $p < 0.01$; **$p < 0.005$; ***$p < 0.001$; ****. Contralateral scrambled peptide vs ipsilateral scrambled peptide $p$-value < 0.0001. Contralateral scrambled peptide vs ipsilateral AP2 inhibitor peptide $p$-value < 0.0001. Contralateral AP2 inhibitor peptide vs ipsilateral scrambled peptide $p < 0.0001$. Contralateral AP2 inhibitor peptide vs ipsilateral AP2 inhibitor peptide $p < 0.0001$. Ipsilateral scrambled peptide vs ipsilateral AP2 inhibitor peptide $p$-value = 0.0004. **d** Recovery curve (Days 1–9) fit to an exponential decay equation. Fitting the recovery curves from **b** reveals that inhibition of endocytosis accelerated the rate of recovery as indicated by a decrease in tau. **e** Graph depicting the thermal sensitivity of the ipsilateral paw in male ($n = 7$; green) and female ($n = 7$; purple) animals that received the scrambled peptide. Gray line represents mean scrambled peptide ipsilateral paw withdraw threshold from **b**. Boxed area corresponds to data point of interest. Data is represented as the mean paw withdrawal ± S.E.M. Boxed area corresponds to 24 h post incision. Significance was determined using a 3-way ANOVA with Fishers Least Significant Difference Post-hoc test $p < 0.5$; *$p < 0.01$; **. Male contralateral scrambled peptide vs female contralateral scrambled peptide $p$-value = 0.0160. **f** Ipsilateral von Frey withdraw thresholds of animals in either the scrambled peptide ($n = 8$) or the AP2 inhibitor peptide ($n = 8$) post-incision. Data is pooled from male and female rats and is represented as mean PWT (as a percentage of baseline PWT) ± S.E.M. Significance determined using repeated measures 2-way ANOVA with Bonferroni correction. Contralateral PWT can be found in Supplementary Fig. 5.

means we employed, should have disrupted membrane homeostasis and negatively impacted membrane-localized receptor signaling (i.e., TrkA), ion channel trafficking, and peptidergic signaling. As a result, animals consistently displayed strong attenuation of pain-like behaviors in models of acute and chronic inflammatory pain (Figs. 1d, 2, 4, and 5, Supplementary movies 1

and 2). The observed effects on von Frey withdrawal thresholds (Fig. 4h) corroborate previously published research implicating peptidergic neurons in the coordination of mechanical and thermal sensitivity during inflammation[7]. Injection of shRNAs into the sciatic nerve may also lead to genetic knockdown of AP2α2 in polymodal nociceptors thus impacting their excitability

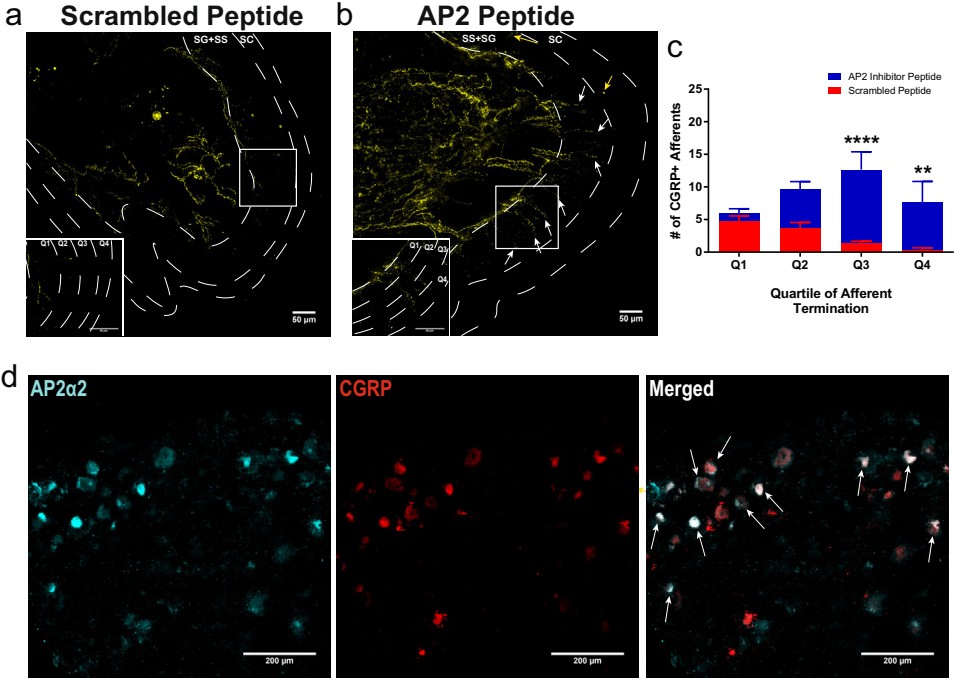

**Fig. 6 Local inhibition of endocytosis in peripheral nociceptors potentiated CGRP immunoreactivity in the superficial layers of the dermis and Ap2α2 is differentially distributed in CGRP+ human DRG neurons. a** Representative image showing CGRP immunoreactivity in an uninflamed hind paw injected with the scrambled peptide. Typically, CGRP immunoreactivity terminates in the proximal stratum granulosum (SG). **b** Representative image showing CGRP immunoreactivity in an uninflamed hind paw injected with the AP2 inhibitor peptide. White arrows: peripheral nerve fibers exhibited robust CGRP immunoreactivity in the very distal layers of the SG and some CGRP immunoreactive fibers could be seen in very superficial stratum corneum (SC) layer. Yellow arrows; peripheral nerve fibers displaying CGRP immunoreactivity in superficial layers of the SC. (Inserts) Magnified sections illustrating SG quadrants. **c** Quantification of CGRP+ afferent termination in each SG quadrant (Q1, Q2, Q3, and Q4; $n = 3$). Significance was determined using a multiple t-test using Holms-Sidak correction with $p < 0.05$; *$p < 0.01$;**$p < 0.005$;***$p < 0.001$;****. Quartile 3 $p$-value = 0.0004. Quartile 4 $p$-value = 0.0076. Data is presented as mean ± S.E.M. **d** Representative immunofluorescent image showing expression patterns of AP2α2 (left) and CGRP (middle) in hDRGs. AP2α2 is differential expressed in CGRP+ DRG neurons. Arrows highlight strong co-localization of CGRP and AP2α2. SG Stratum Granulosum, SS Stratum Spinosum, SC Stratum Corneum.

during pain resulting in a decrease in lifting behaviors following shRNA knockdown (Fig. 1e) that is not apparent after local AP2 inhibitor peptide injection (Fig. 3a). These effects may also be the result of alterations in synaptic transmission since AP2α2 is not fully excluded from the presynaptic interface[15], which would clarify why there was no observable differences between male and female mice in the shRNA knockdown experiments (Fig. 2). Nevertheless, the magnitude of our effects on von Frey withdrawal thresholds suggests that CGRP release from peripheral neurons indirectly contributes to the development of mechanical sensitivity.

Furthermore, accumulation of membrane-localized $K_{Na}$ channels may also contribute to the observed changes in pain-like behaviors. Gururaj et al.[32] provided evidence that inhibition of neuronal endocytosis resulted in the membrane retention of large-conductance Kcnt1 (Slack) $K_{Na}$ channels, which caused the lack of PKA-induced hyperexcitability in cultured DRG neurons. In acutely dissociated neurons from AP2α2 in vivo knockdown, we also found a lack of PKA-induced hyperexcitability (Fig. 1c). Persistent exocytosis of LDCVs without accompanying endocytosis could also cause an increase in membrane Kcnt2 (Slick)[55] channels, another large-conductance $K_{Na}$ channel that was shown localized to CGRP containing LDCVs. We observed a significant reduction in CFA-induced thermal sensitivity after AP2 inhibitor peptide injection (Fig. 4b), indicating that blocking ongoing endocytosis, even after full-blown neurogenic inflammation (Supplementary Fig. 8), altered neuronal excitability. Indeed,

overexpression of Kcnt2 channels in DRG neurons was shown to blunt action potential formation[55].

Using an antigenic lipidated peptidomimetic (HA-peptide) we showed molecular partitioning in both non-inflammatory (Fig. 3a) and inflammatory (Fig. 3b) conditions. We used the HA-peptide as a proxy to understand how small lipidated peptides penetrate into neuronal afferent endings. Both conditions showed HA-immunoreactivity in the dermis, whereas the epidermis and muscle tissue appeared devoid of signal. These findings suggest that either the lipidated peptide was rapidly cleared from these compartments in the hind paw, or the hydrophilic extracellular matrix prevented peptide penetration. Behavioral testing showed that after a single injection, lipidated peptides have a longevity in vivo that was similar to the one observed for the HA-peptide in vitro (Supplementary Fig. 3). The longevity of small lipidated peptides in cells might depend on membrane turnover kinetics[56]. Injection of our lipidated AP2 inhibitor peptide is aligned with current clinical administration of other FDA-approved lipidated peptides. For example, dulaglutide (Trulicity®) and semaglutide (Ozempic®) are different lipidated Glucagon-like peptide 1 (GLP-1) peptides subcutaneously injected (sometimes daily) to treat diabetes. The GLP-1 peptide (~30 amino acids) is considerably larger than the peptides described here. Also, in order to achieve systemic absorption and prolonged stability the GLP-1 peptide is administered at 100–1000-fold higher than the concentrations of the lipidated peptides we administered to rodents. We envision locally targeting peripheral

nerve afferents at doses that would have minimal systemic absorption. However, dosing of both AP2 inhibitor peptides and peptides that target Na$_V$1.8 channels (Supplementary Fig. 6) require further exploration.

When the sexes were pooled together, we observed robust, significant reductions in pain-like behavior by the AP2 inhibitor peptide (Figs. 4b and 5b) that lasts for many days. When the data was segregated by gender, we uncovered sex-dependent pain-like behaviors in two different models of inflammatory pain across two species of animals. These results could only have been observed because of the long-lasting pharmacological inhibition of CGRP[+] nociceptors and the implementation of an exponential decay best-fit model to interpret thermal responsiveness. This allowed us to characterize the efficacy of our AP2 inhibitor peptide and quantify recovery kinetics. However, the sex-dependent difference was contingent on whether the AP2 inhibitor peptide was given before or after development of inflammation. Post-facto injection of the AP2 inhibitor peptide resulted in rapid recovery of thermal sensitivity in male mice, whereas females showed a delayed response to the peptide (Fig. 4e, Supplementary Fig. 4C, H). Additional analysis showed that the AP2 inhibitor peptide produced an analgesic-like effect in male and female mice following injection, as indicated by an increase in dynamic weight borne on the ipsilateral paw 48-hours post peptide injection. The use of the Dynamic Weight Bearing apparatus allowed us to observe changes in coping behavior governed by normally silent TRPV1 + nociceptors that are not readily accessible in Hargreaves and von Frey assays[39,57]. Moreover, there was a profound difference in $\tau$ – values when comparing the male scrambled peptide group to the male AP2 inhibitor peptide groups. This effect was not observed within the female group, however. In this sense, inhibition of peripheral endocytosis results in rapid recovery, and profound analgesia in male mice, whereas female mice experience a significant analgesic effect, yet the rate of thermal recovery does not appreciably change. In post-incisional models of pain, prior research found a lack of sex-differences in mechanical sensitivity[58,59] and hot plate assessment[49]. By using the Hargreaves's method to assess discreet unilateral thermal responsiveness, we observed a sexually dimorphic pain response. In this model, however, a 6-h hind paw pre-injection procedure was included (Fig. 5a) so observed changes in pain-like behavior would be a result of both pre-emptive and post-facto afferent modulation. Male animals from both control and experimental groups displayed linear recovery profiles, similar to previous iterations of this pain model[34,60]. Although male rats did not display substantial changes in behavior as measured by traditional methods (Supplementary Figs. 5A, B), there was a large difference in the rate of recovery for males that received the AP2 inhibitor peptide (Supplementary Fig. 5C). Conversely, female rats responded well to the AP2 inhibitor peptide in all three measures of analgesia (Supplementary Fig. 5D–F). Of note, the control female animals' recovery curve appeared biphasic, characterized by two peaks that occur on day-2 and again on day-7, (Supplementary Fig. 5D) which we predict would be better fit by a two-phase decay equation indicative of sequential fast and slow neuroplastic processes.

In both inflammatory pain models, we have observed a sexually dimorphic response to both injury and our peptidomimetic suggesting that males experience a more complete analgesia, compared to females, with peripherally acting compounds after inflammation is established (as measured by absolute thermal thresholds, A.U.C. and analysis of recovery). In a situation where pre-emptive analgesia is desired, females appear to respond better than males. Altogether, these observations might be the manifestation of a fundamental differences in male and female nociception physiology, and by extension pathophysiology, during the

initiation and maintenance of inflammatory pain. Since our peptidomimetic is peripherally acting, it would stand to reason that nociception and pain-signaling is dominated by peripheral mechanisms in males[61] and inhibition precipitates a rapid and robust effect. Whereas females have equal contributions from both peripheral and central nervous systems, therefore, inhibition of peripheral nociceptors results in a temporally slower onset of analgesia. This is supported by sex-differences in the CGRP receptor components within trigeminal ganglion, medulla[62], and the spinal cord[63] despite equivalent levels of CGRP mRNA. Through the long-term inhibition of CGRP[+] nociceptors, we have unexpectedly uncovered sex differences in how thermal hyperalgesia is manifested and recovers. Our data is consistent with the idea that there are sex-differences in both the prevalence and the intensity of chronic inflammatory[64] and post-operative pain in humans[65].

Local administration of therapeutics targeting peripheral nociceptor afferents is becoming a more preferable approach to treat pain because it decreases side effects, including addiction[66–68]. For example, local injection of reformulated anesthetics is a current alternative to opioids use for pain relief[69] although their efficacy is controversial[70]. Nonetheless, two major challenges remain for locally applied drugs: specificity and duration of action. For patients with ongoing injury-associated pain and associated inflammation, targeting specifically the TRPV1/CGRP[+] class of afferent fibers may be key in providing effective pain relief[26]. Nevertheless, we have demonstrated locally administered lipidated peptidomimetics are able to produce specific and long-lasting reductions in pain-like behaviors.

## Methods

**Animals.** All animals were purchased from Envigo and age/weight matched for all experiments. All animals used were housed in the Laboratory Animal Facilities located at the University at Buffalo (UB) Jacobs School of Medicine and Biomedical Sciences on a 12-hour light/dark cycle. For consistency, all animals were singly housed for the duration of the experiments and all pain assays were conducted during the light phase. All animals were given access to food and water ad libitum. All animal experimentation was conducted in accordance with the guidelines set by the Guide for the Care and Use of Laboratory Animals provided by the National Institute of Health. All animal protocols were reviewed and approved by the UB Institute Animal Care Use Committee.

**In-vivo transfection of sciatic nerves with α2 targeted shRNAs and in vivo-jetPEI**®. C57BL/6 mice were anesthetized with 2% isoflurane until reflexes to hind paw and tail pinching subsided. Then optical lubricant was placed in each eye and isoflurane concentration dropped to 1% for the duration of the procedure. The animal was placed in a prone position to allow for access to the paraspinal area. The area was shaved using electrical sheers and the area was cleaned first using chlorohexidine, followed by 70% ethanol, and finally sufficient amounts of iodine (all disinfectants were provided by UB Laboratory Animal Facilities). After disinfection, a 3 cm posterior longitudinal incision is made at the lumbar segment of the spine. Utilizing sterile toothpicks, ipsilateral paraspinal muscle was carefully separated to expose the sciatic nerve. Using autoclaved sticks, the nerve was manipulated slightly to ease injection. 1.5 uL of PEI/shRNA plasmid DNA polyplexes, at an N/P ratio of 8, was injected directly into the right sciatic nerve using a syringe connected to a 32-gauge needle (Hamilton 80030, Hamilton, Reno, NV). AP2α2 shRNAs and control shRNA were purchased from Santa Cruz Biotechnology (Santa Cruz, CA, USA). Following injection, the needle was maintained in the sciatic nerve for at least 1 min to promote diffusion of the polyplexes. The wound was closed with wound clips and mice were post surgically observed to ensure no adverse effects due to the injection. Mice were given 7 days of recovery before behavioral testing resumed.

**Myristoylated peptide preparation.** Sequences of peptides used in the study can be found in Supplementary Table 1. Lipidated peptidomimetics were initially dissolved in 10 μL of DMSO to create a working stock solution. Appropriate volumes of the DMSO stock solution were dissolved in 1 mL of sterile saline to generate 100 μM aliquots for future testing. Final DMSO concentration was <0.05%. These aliquots alongside any stock solutions were frozen at −80 °C until needed, at which point one aliquot was thawed, injected, then discarded to minimize freeze–thaw cycles of samples.

**Formalin assay**. Male and female C57BL/6 mice were randomly assigned to either control or experimental groups. Animals received a 20 μL intraplantar injection of 100 μM (3.154 μg total) lipidated peptidomimetic 24 h prior to experimentation. Animals were habituated to the formalin testing chamber for 30 min or until exploratory behavior ceased the day of experimentation. Following the habituation period, animals were removed from the chamber and given an (20 μL) intraplantar injection of 5% formalin into the ipsilateral hind paw, then immediately placed back into the testing chamber and recorded. Animals were recorded for at least 90 min after formalin injection using Active WebCam software, version 11.6 (PY Software). Videos were subsequently scored for number of paw licks, number of paw lifts, and number of full body flinches. All behaviors were scored for a full minute, every five minutes, for 90 min of video recording. Scorers were blinded to experimental conditions.

**Complete Freund's adjuvant induced inflammatory pain**. Male and female C57BL/6 mice were randomized into experimental and control groups. In order to maintain consistency with regard to site of injection, mice were anesthetized and injected with a 32-gauge disposable syringe filled with 20 μL of Imject™ Complete Freund's Adjuvant (Thermo Fisher Scientific) into the plantar surface of the right hind paw and allowed to recover. Behavior testing resumed 24 h post-CFA injection at which point the animals received a 20 μL intraplantar injection of 100 μM (3.154 μg total) lipidated peptidomimetic immediately after the conclusion of day 1 of behavioral testing. In order to minimize experimental error between groups, each group of animals received CFA from previously unopened, vacuum sealed glass ampules ensuring CFA of identical specific activity.

**Incisional post-operative pain model**. To model post-operative pain, an established rat incisional model was used[41]. In short, male and female rats were randomized into either experimental or control groups. On the day of surgery, the animals were anesthetized and placed into a prone position. Once the animal was under a surgical plane of anesthesia, a 200 μL intraplantar injection of 100 μM (31.54 μg total) lipidated peptidomimetic was made into the ipsilateral hind paw. Afterwards, the animals were returned to their home cage and allowed to recover. On the same day, 6 h after the pre-injection, the animals were anaesthetized, placed into a prone position, and prepared for incision injury. The ipsilateral hind paw was sterilized using successive swabs of chlorhexidine, 70% ethanol, and iodine. Then, using a size 10 scalpel, a 1 cm long incision was made into the plantar surface of the ipsilateral hind paw. Short, yet firm, strokes were used to make incisions through the skin, fascia, and muscle of the hind paw. Following incision, two 50 μL injections, containing 100 μM (7.885 μg per injection) of the lipidated peptidomimetic, were made into each half of the incised plantar muscle. Following injection into the muscle, the skin was sutured using 6/0 silk sutures (Ethicon) in a continuous manner to discourage removal of sutures. Upon conclusion of suturing, four 25 μL injections containing 100 μM (3.9425 μg per injection) of the lipidated peptidomimetic, were made into a quadrant adjacent to the incision. Finally, the animals were returned to their home cage and allowed to recover for at least 16 h.

**Thermal sensitivity testing**. Prior to testing, animals were allowed to habituate to the testing room for 1 h on each day. Animals were placed on an enclosed elevated frosted glass platform (Ugo Basile) and allowed 30 min for habituation. Once exploratory behavior ceased, an automatic Hargreaves apparatus was maneuvered (Ugo Basile) underneath the hind paw(s) of the animals. Paw withdrawal latency was calculated as the average of four trials per hind limb. Each trial was followed by a 5-min latency period to allow adequate recovery time between trials.

**Mechanical sensitivity testing**. Each day, animals were placed on an enclosed elevated wire-mesh platform (Ugo Basile) and allowed 1 h to habituate to their enclosure. For mice, Touch Test® Sensory Probes (Stoelting) were applied to the plantar surface of the contralateral and ipsilateral hind paw. Filaments were applied in an ascending or descending order following the Simplified Up-Down method (SUDO) for von Frey assays[71]. In short, the middle filament of the series was presented to the hind paw of the animal. If a response was elicited, the next lowest filament in the series was presented. If no response was elicited, the next highest filament in the series was presented. This method of filament presentation was repeated 5 times, with the 5th filament presentation being the last one. Then an adjustment factor was added to the filament value and the force of paw withdrawal was calculated utilizing a series of conversion equations. Each paw per animal was given a 5-min latency period between filament presentations to reduce the chance of sensitization in the paw.

Mechanical sensitivity testing on rats was conducted using an automated Dynamic Plantar Aesthesiometer (Ugo Basile). Rats were placed in an elevated enclosure atop a wire mesh platform. On each testing day, rats were given 1 h to habituate to the room and the chamber. Mechanical sensitivity thresholds were determined using an automatic probe affixed with a mirror. The probe was set to exert a maximum upward force of 50 g over a span of 20 s. The force necessary to elicit a response (as measured by swift removal of the paw from the probe) was recorded and this was repeated 5 times per hind paw with at least 5 min in between recordings to minimize sensitization.

**Dynamic weight bearing**. Animals were allowed to habituate to the testing room for 30 min before initiating testing. Following habituation, animals were weighed and then placed into a mouse Dynamic Weight Bearing (DWB) enclosure (BIO-SEB). The enclosure is connected to a weight sensor and a video camera that allows for automatic acquisition and scoring of weight bearing behavior. Animals were given a 180 s latency period before weight acquisition to allow for exploration immediately followed by a 300 s acquisition period. At the conclusion of the experiment, acquisitions were manually validated using the BIOSEB DWB-2 software, version 2.0.63, to ensure accuracy of readings. Automatically scored postural data was manually validated until at least 2.5 min of the acquisition video was manually scored. Finally, the data was exported to Microsoft Excel, version 18.2106.12410.0, (Microsoft) where it was compiled and formatted for import into Prism version 6, for most of the statistical analysis, and Prism version 9, for 3-way ANOVA calculation (Graph Pad).

**Immunofluorescent staining**. Animal tissue was collected following a standard transcardial perfusion protocol, as previously described[72]. Slices for staining were made at 15 microns for DRGs (mouse and human), and 50 microns for the hind paws. Mouse DRG (mDRG) tissue were affixed to charged Superfrost microscope slides (Fisherbrand). The sections were first washed 3 times with PBS, then incubated overnight in blocking media (10% Normal Goat Serum, 3% Bovine Serum Albumin, and 0.025% Triton X-100 in PBS). The next days, the slides were incubated, overnight, in primary antibodies (Mouse anti-CGRP; 1:500 Abcam, Rabbit anti-AP2α2 1:500 Abcam[20]). The next day, the slides were incubated with the secondary antibodies (Goat anti-rabbit 546 1:1000 Invitrogen, Donkey anti-Mouse 488 Abcam). The following day, the slides were rinsed 3 times with PBS and incubated with an IB4-647 conjugate (Invitrogen) at room temperature for 2 h. Afterwards, the slides were rinsed twice more and mounted using ProLong™ Glass Antifade Mountant (Invitrogen).

Human L5 dorsal root ganglia (hDRGs) were purchased from Anabios. The donor was 49 years old, female, and had unremarkable past medical history. The study was certified as exempt by the University at Buffalo Internal Review Board because the hDRGs were collected from a donor and no identifying information was shared with the researchers. The hDRGs were initially preserved in formaldehyde and shipped on dry ice in 70% ethanol. Upon arrival, the hDRGs were rehydrated, sequentially, in decreasing ratios of PBS to water: 24 h in 50% PBS then 24 h in 30% PBS. Following rehydration, the hDRGs were cryoprotected in 30% sucrose at 4 °C, and submerged in tissue freezing media (Electron Microscopy Sciences) and frozen using dry-ice chilled 2-methylbutane. Once the resulting blocks were thoroughly frozen, they were placed into a −80 °C freezer for 48 h. Cryosections were taken and mounted onto charged Superfrost microscope slides. hDRGs were sectioned and stained in a similar manner described above for the mDRGs using the same antibody concentrations.

Hind paws were stained as free-floating sections and probed in a similar manner described for DRG tissue. The following primary antibodies were used where applicable: mouse anti-HA primary antibody (1:500 Abcam) and mouse anti-CGRP (1:500 Abcam). The secondary antibody used in both instances was a goat anti-mouse 555 secondary antibody (1:1000 Abcam). After washing the secondary antibody, the sections were incubated in increasing amounts of thiodiethanol (TDE). TDE acts a tissue clearing agent aiding in fluorescent signal penetration[73]. The first incubation consisted of 10% TDE in a 1:1 solution of PBS in ddH$_2$O overnight. The second incubation was in 25% TDE in 1:1 PBS in ddH$_2$O overnight. The third incubation was in 50% TDE in 1:1 PBS in ddH$_2$O overnight. The final incubation was in 97% TDE in 1:1 PBS in ddH$_2$O. Following the final TDE immersion, the sections were rinsed once with 1:1 PBS in ddH$_2$O and mounted onto charged Superfrost microscope slides using ProLong™ Glass Antifade Mountant.

All slides were allowed 24 h to set at 4 °C, before imaging. All images were acquired using a Leica DMi8 inverted fluorescent microscope equipped with a sCMOS Leica camera (Lieca) and connected to a HP Z4 G4 Workstation (HP) loaded with THUNDER enabled LAS X imaging software, version 3.7.3.23245 (Lieca). All images were analyzed using a separate HP Z4 G4 workstation that was loaded with the LAS X imaging software. Images were exported and further modified (i.e., addition of scale bars, heat-map transformations) using ImageJ, version 1.53c, (NIH) and compiled into files using Adobe Illustrator version 25.4.1 64-bit (Adobe).

**Histology**. To collect hind paws for histology, animals underwent the previously described incisional pain protocol. However, these animals did not undergo behavioral testing. Hind paws were collected following transcardial perfusion. The muscle, fascia, and skin were carefully separated from the bone and post fixed in paraformaldehyde for 24 h. Following post-fix, the hind paws were stored in 70% ethanol and transported to the Histology core housed in the Jacobs School of Medicine, SUNY at Buffalo where the hind paws were sliced at 20 microns, processed, stained with eosin and hematoxylin, and mounted onto slides and cover slipped. These processed samples were then returned to the researchers and imaged on a Leica DM 6B upright microscope equipped with a sCMOS Leica camera (Lieca) and connected to a HP Z4 G4 Workstation (HP) loaded with THUNDER enabled LAS X imaging software.

**Electrophysiology**. Glass electrodes were pulled using a vertical pipette puller (Narishige Group) and fire-polished for resistances of 5–8 MΩ. Current–clamp recordings were performed on dissociated adult DRG neurons from mice in vivo transfected with either scrambled control shRNA or α2 targeted shRNAs. To collect adult mouse DRGs for culture, adult mice were euthanized with $CO_2$ and decapitated. The spine was isolated and placed in a Petri dish with Hanks Balanced Salt Solution (Corning). DRGs were dissected and dissociated in 0.28 Wunsch units/mL Liberase Blendzyme (Roche) for 1 h at 37 °C. The dissociated neurons were then plated on glass coverslips coated with poly-d-lysine (PDL) laminin and cultured in Neurobasal-A Medium (Invitrogen) supplemented with B-27 Supplement (Invitrogen), and nerve growth factor (NGF). To stimulate PKA, dissociated neurons were incubated with the cell-permeating cAMP analog Sp-cAMPs (50 μM) for 30 min at 37 °C before recording. Dissociated neurons were incubated with Alexa fluor-488 conjugated IB4 (Invitrogen I21411) for 5 min, washed thrice with sterile PBS before recordings began. Only non-fluorescing small- and medium-sized DRG neurons were recorded. Firing frequency was examined by injecting a supra threshold stimulus of 400 pA for 1000 ms. A pipette solution consisting of 124 mM potassium gluconate, 2 mM $MgCl_2$, 13.2 mM NaCl, 1 mM EGTA, 10 mM HEPES, pH 7.2 was used. A bath solution consisting of 140 mM NaCl, 5.4 mM KCl, 1 mM $CaCl_2$, 1 mM $MgCl_2$, 15.6 mM HEPES, and 10 mM glucose, pH 7.4 was used. All data were acquired using Multiclamp-700B (Molecular Devices), digitized, and filtered at 2 kHz. Data acquisition was monitored and controlled using pClamp 10.2 and analyzed using Clampex (Molecular Devices).

**Western blot analysis**. Total protein was collected from DRG tissue collected from animals following experimentation. DRGs were homogenized in chilled RIPA buffer containing a protease inhibitor (Sigma) and stored at −80 °C until needed. All samples were run on Mini-PROTEAN TGX Precast Gel (Bio-Rad) and transferred to a 0.45 μm nitrocellulose membrane (BioRad). Membranes were probed overnight at 4 °C with rabbit anti-AP2α2 (1:1000, Abcam) or rabbit anti-Actin (1:1000, Sigma) in 5% bovine serum albumin (BSA) prepared in 1x tris-buffered saline-tween (TBST). On the following day, membranes were washed three times for 5 min in 1x TBST before being incubated for 1 h at room temperature in a secondary anti-rabbit horseradish peroxidase conjugate antibody (1:5000; Promega) prepared in a 5% BSA in 1x TBST solution. After secondary anti-body incubation, the membrane was washed three more times for 5 min per wash before being developed and imaged. Bands were visualized with enhanced chemiluminescence on a Chemidoc Touch Imaging System (Bio-rad) and quantified with Image J Software (NIH). Each experiment was repeated at least three times.

**Statistics**. All statistical tests were performed using Prism (GraphPad). The data are shown as means ± S.E.M. Power analysis was conducted for animal experiments to achieve detection limits with an $\alpha$-value set at 0.05. Statistical significance was determined utilizing a $p$-value <0.05 for all experiments. Repeated measures two-way ANOVA statistical tests with multiple comparisons and stringent Bonferroni correction, one-way ANOVA with Holms-Sidak correction, and Student's $t$ test were used where appropriate. Tau analysis was conducted using the following equation: $W(t) = (W_0 - p)e^{-kt+p}$, where $W_t$ is the withdrawal threshold at given time $t$, $W_0$ is the withdrawal threshold at $t = 0$, $p$ is the plateau value, $k$ is the rate constant, and $t$ is the time in days. Constraints were implemented to prevent near infinite tau values; $W_0 > 1$ and $p < 16$.

**Reporting summary**. Further information on research design is available in the Nature Research Reporting Summary linked to this article.

## Data availability
All data generated in this study are provided in the Source data file. Source data are provided with this paper.

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

## Acknowledgements

We thank Dr. Wade Sigurdson for assistance with microscopy for the immuno-fluorescence experiments. We thank Dr. Elsa Daurignac for critical reading of this manuscript. This work was supported by the National Institute of Health Grants NS108087, NS113991, and Buffalo Accelerator Fund (all to A.B.). Further support was provided by GM095459 (R.P.) and a Diversity Supplement Award to NS108087.

## Author contributions

R.P. performed all the biochemical, electrophysiological analyses, and the associated behaviors during in vivo AP2α2 knockdown. R.P. performed the initial immunohisto-chemical analysis of AP2α2 in mouse DRG neurons, the entire analysis in human DRG neurons and conducted all the dermal immunohistochemical analyses involving the HA peptide and CGRP. R.P. conducted all the behaviors associated with AP2 inhibitor peptide and performed all the data analysis. V.Y. conducted AP2α2 knockdown and performed immunohistochemical analysis. V.Y. performed the mechanical responsive-ness experiments during incisional pain in rats. K.D.P. piloted the AP2α2 knockdown studies assessed by formalin assay. G.D.S. conceived the lipidated HA peptide approach and assisted on neuronal dissociation. K.B. and A.A. conducted the incisional pain studies assessing the analgesic effects of Na$_V$1.8 targeted peptides. K.B. and A.A. scored all the formalin assays. The manuscript was written by R.P. and A.B. All authors critically revised and gave final approval to this manuscript.

## Competing interests

A. Bhattacharjee is the co-founder of Channavix Inc., a company developing non-opioids drugs for pain. A patent has been filed on the use of lipidated peptidomimetics targeting endocytosis to treat pain. All other authors declare no competing interests.
