## [Peer Review File · Nature Communications]

Inhibiting endocytosis in CGRP+ nociceptors attenuates inflammatory pain-like behaviorREVIEWER COMMENTS

Reviewer #1 (Remarks to the Author):

This is a very interesting study that asks whether disrupting membrane trafficking could have an analgesic effect in rodent pain models. The approach used a peptide inhibitor of the clathrin-associated adapter protein AP2a2. By focusing on AP2a2 instead of the $\alpha 1$ isoform, this presumably disrupted targeting of non-synaptic endocytosis, which is important since this would mostly affect modulatory peptides (and ion channels) as opposed to synaptic transmission by neurotransmitters. This is clever. In addition, another strength is the use several pain paradigms and showing colocalization of AP2a2 and CGRP in rodent and human DRGs. Finally, the translational strength of using local application in the periphery is clear since this will localize the effects and likely reduce risks of adverse side effects. However, there are some major concerns that should be considered.

1. A major concern is that the study relies on nociceptive withdrawal reflex assays and to a lesser extent on nocifensive behaviors. Study would be greatly strengthened by using operant based pain assays for at least one of the pain models to increase likelihood of translation to humans.
2. A second major concern is with the data reporting sex difference of AP2 inhibitory peptide treatments, which may be overstated. The difference is not consistently seen, which may reflect the small sample sizes, and when seen is not that impressive, which raises questions of biological relevance. In Fig 4, data are taken from a single experiment with only $n=4$ mice. Mouse behavior is too variable to draw any conclusion from just 4 mice. Further, the tau calculation mainly points out a difference in the control values between male and female (both decrease but from different starting points). The same comments apply to Fig. 5 for which they used rat data ($n=6$ for each sex) and appear to infer that the peptide worked better in females than males, although this is not clear in the Discussion. A concern is that there apparently was not any sex dependence of the peptide in mice since only the scrambled controls for mice are shown in suppl fig 6, and in this experiment, the scrambled controls do not differ between sexes, unlike in rats. Taken together, it seems that there is variability in the assays and when the groups are small ($n=4, 6$), differences are being observed that really need to be confirmed with larger n 's and, importantly, in independent experiments.
3. Hard to get a grasp on prevalence of AP2a2 in DRG from data in fig 1. Panel A looks like a small percentage, while panel B looks like almost all neurons.
4. The AUC data should be represented as individual data points (e.g. as done in Fig 4A) not as traditional bar and plungers (e.g. Fig 4E) to give better appreciation of the spread in each group.
5. Were there sex differences with the shRNA experiments? If not, this should be addressed, pertinent to point #2.
6. How explain differences between shRNA and the inhibitory peptide? Eg shRNA affects lifting behavior, mechanical sensitivity, etc, while peptide did not.
7. The point of testing other peptides in suppl fig 7 is not clear. For example, what was the rationale for selecting these peptides?

Minor

1. Need to say what AP2a2 is in the abstract and abstract should clearly state the gender differences if that is kept in the paper.
2. Fig 1F, while nice to have representative images and videos, data should be quantified for presentation.
3. Fig 2 B is not percent of baseline as labeled for the Y axis. It is a ratio, as correctly labeled in parentheses.

Reviewer #2 (Remarks to the Author):

This manuscript reports that AP2a2-mediated endocytosis in CGRP-expressing nociceptors contributes to acute and chronic inflammatory pain in mice and rats. AP2a2 was found to be preferentially expressed in CGRP+ve nociceptors in mouse and human DRG. Knockdown of AP2a2

attenuated PKA-evoked nociceptor excitability and blunted acute nociceptive behavior in mice. Lipidated probes incorporated into membranes of CHO cells and DRG neurons, where they were retained for 72 h. Lipidated peptide-based inhibitors of AP2a2 suppressed acute nociceptive behavior and reversed CFA-evoked inflammatory nociception in mice. They also suppressed surgically-induced nociception in rats. These anti-nociceptive actions lasted for several days. Injection of the lipidated AP2a2 inhibitor led to increased CGRP immunoreactivity in distal layers of the stratum granulosum, suggesting that inhibition of endocytosis suppresses CGRP release in the periphery.

This is an interesting manuscript that builds on the authors' published work showing that AP2 clathrin-mediated endocytosis underlies DRG neuronal sensitization through effects on trafficking of sodium-activated potassium channels. The present study extends this work by examining nociception in animal models. The most striking finding is that lipidated AP2a2 inhibitors have sustained anti-nociception actions in mice and rats. The authors use multiple approaches to support their conclusions. In general, the experiments are clearly described and the results are convincing.

Several issues require attention.

1. Some experimental approaches are not fully described.

a) Fig. 1D shows the effects of AP2a2 shRNA on PKA-induced hyperexcitability of DRG neurons.

How was PKA activated in these experiments?

b) Fig. 2 shows the effects of AP2a2 shRNA given pre- or post-inflammation. Please clarify the timing of shRNA administration relative to the CFA injection since this is not clearly stated in the methods.

2. Some results require more complete quantitation.

a) Fig. 1A shows preferential localization of AP2a2 in CGRP+ve DRG neurons. The results would be more convincing if the authors quantify the proportions of CGRP+ve and IB4+ve neurons that express AP2a2 in several animals?

b) Fig. 1D shows representative traces of the effects of AP2a2 shRNA on PKA-induced hyperexcitability of DRG neurons. Pooled data of action potential firing should be shown and analyzed statistically.

c) Can the authors quantify the effects of AP2a2 inhibitors on CGRP concentrations measured by ELISA rather than rely on semi-quantitative IF?

3. The authors examined the distribution of lipidated probes by localizing a lipidated HA peptide. They extrapolated these findings to the distribution of lipidated AP2a2 inhibitors. However, this may not be appropriate since the AP2a2 inhibitor may preferentially target AP2a2 in CGRP+ve neurons. Can experiments be designed to localize the lipidated AP2a2 inhibitors?

4. Fig. 3 shows uptake of lipidated HA probes into afferent fibers. What is the evidence that these are afferent nerve fibers? Is it possible to co-label tissues with a neuronal marker? Do the authors propose preferential probe uptake into neurons and, if so, how does this occur?

5. Supplemental Fig. 3 claims to show retention of lipidated HA probes in the plasma membrane of CHO cells and DRG neurons for several days. However, the images are low magnification/resolution and the entire cell is labeled. Higher power confocal images are required to demonstrate that the probe is confined to the plasma membrane. Given the rapid turnover of membranes, does the probe traffic to and from the plasma membrane and endosomes? Panel A lacks a scale bar. A control for selectivity of the HA antibody is required.

6. The authors propose that defective CGRP release from the peripheral endings of nociceptors might explain the anti-nociceptive properties of AP2a2 knockdown and inhibition. This conclusion is based on increased CGRP staining in peripheral tissues in animals treated with the lipidated AP2a2 inhibitor. Given that CGRP along with Substance P (SP) is a major mediator of neurogenic inflammation in the periphery, how do the authors explain that the AP2a2 inhibitors do not affect paw inflammation? Are SP levels similarly affected? The central release of CGRP in the spinal cord mediates pain transmission. Are there changes in CGRP levels in the central projections of nociceptors in animals treated with AP2a2 inhibitor?

7. In the discussion, the authors also raise the possibility that altered trafficking of ion channels

might also explain the anti-nociceptive actions of AP2a2 inhibitors. This is based on their past work. There are several reports that inhibitors of clathrin and dynamin can inhibit nociceptor hyperexcitability and nociception in rodent models. These effects have been attributed to inhibition of endosomal signaling of GPCRs – CLR, NK1R, PAR2. The authors may wish to speculate that AP2a2 inhibitors could also block endosomal signaling of GPCRs that mediate pain transmission.

8. The studies of human nociceptors add translational relevance but are limited because they are of nociceptors from a single donor. Ideally, observations would be replicated in DRG from several subjects. The proportion of different types of neurons expressing AP2a2 could then be quantified.

Reviewer #3 (Remarks to the Author):

This manuscript entitled "Inhibiting endocytosis in CGRP+ nociceptors as a treatment for inflammatory pain" by Powell et al. examine local disruption of nociceptor endocytosis and use various inflammatory pain models to characterize the in vivo contribution of extra-synaptic AP2 clathrin-mediated endocytosis (AP2-CME) to inflammatory pain.

Overall, the data presented do not provide further evidence for "peptidergic nociceptors as executive regulators of inflammatory pain" or support "nociceptor endocytosis as a promising target for local, specific, and long-lasting treatment of inflammatory pain." The experiments described here are not truly validating the proposed target in inflammatory pain. The in vivo data supporting the claims in the manuscript are problematic:

Figure 1.

(1) Panel C (right): Is the blue bar that is labelled as Control, the scrambled shRNA or a different Control?

(2) Panel D: Data for PKA stimulation under control conditions is missing, to enable interpreting the data with the scrambled and AP2 shRNA in perspective.

(3) Panel E: A baseline 5% formalin response cohort is not shown, making it hard to appropriately compare the results with either the scrambled or AP2a2 shRNA groups.

(4) The number of licks, lifts and flinches following 5% formalin are quite high with the scrambled peptide for an N=6 and is hard to put into perspective without having a control group to understand the true baseline. How exactly were the behaviors scored from the video? Are these male or female mice, or is this pooled data? This is not clarified in the legend.

Figure 2.

(1) Absence of baseline CFA response in animals to compare with the scrambled (n = 8) and AP2a2 (n = 8) shRNA groups. Are these pooled male and female?

(2) Negligible difference in thermal sensitivity, in the AP2a2 group (less than 3 sec) and the AP2a2 is still significantly sensitive to thermal stimuli at day 8.

(3) The mechanical sensitivity change is also not particularly impressive. Data are better plotted as withdrawal threshold instead of mean percent of baseline to understand a true response.

(4) A 5 sec withdrawal latency in the scrambled shRNA group and a 2 second change in the AP2a2 shRNA makes it hard to interpret the results.

Figure 4.

(1) Panel A: A control group is missing to enable appropriate comparison to either the scrambled or

AP2a2 shRNA groups. It is odd that liking behavior shows a difference between the scrambled and AP2a2 groups but lifting and flinching do not show a difference. Are these pooled male and female rats?

(2) A two second difference in data presented in panels C and D with an N=4 is hard to justify as an analgesic response. Thermal sensitivity in the CFA model is generally variable across time. Hence, it is difficult to understand how such minimal variability was achieved with an N=4 in the experiments presented. Given the minimal response observed in data presented in panels C and D, it is not clear whether the additional analysis presented in panels E-K are truly valid or provide an over interpretation of the response to the AP2shRNA. Panel K shows a difference between the two groups only on Day 2.

(3) In both Figures 2 and 4 it is not clear as to whether the thermal sensitivity was measured at the same time on each day.

Figure 5.

(1) Panel D- it is not clear that the response in the ipsilateral AP2a2 group has a late phase and an early phase. Is the statistical comparison to the sham ipsilateral group? Again, the data here is hard to interpret in the absence of a control group with no surgery.

(2) In these experiments, the treatment paradigm of injecting peptide into the muscle then suturing and injecting more peptide after suturing is a bit odd.

Supplementary Figure 7.

(1) Panel A: it is hard to interpret the effects with P1-P4 in the formalin model. The AP2a2 group (n=6) shows no decrease in the lifting behavior compared to the scrambled peptide, but the P1-P4 groups (n=3) reduce lifting behavior, but not licking behavior or flinching behavior.

(2) It is odd that the response of the scrambled group and the AP2a2 group are remarkably similar to that in Figure 1.

(3) In Panel B, are the responses to ipsilateral Ch1001 and Ch1002 being compared to the contralateral Ch1001 and Ch1002, or to the scrambled or the AP2a2 group?

Overall, it is not clear how the proposed mechanism supports the therapeutic possibility, given the negligible effects observed overall in the 3 pain models. It does not seem that the experiments described here are truly validating the proposed hypothesis and target.

We thank the reviewers for their guiding criticisms. Below are our responses to their concerns.

Reviewer #1 (Remarks to the Author):

This is a very interesting study that asks whether disrupting membrane trafficking could have an analgesic effect in rodent pain models. The approach used a peptide inhibitor of the clathrin-associated adapter protein AP2a2. By focusing on AP2a2 instead of the $\alpha 1$ isoform, this presumably disrupted targeting of non-synaptic endocytosis, which is important since this would mostly affect modulatory peptides (and ion channels) as opposed to synaptic transmission by neurotransmitters. This is clever. In addition, another strength is the use several pain paradigms and showing colocalization of AP2a2 and CGRP in rodent and human DRGs. Finally, the translational strength of using local application in the periphery is clear since this will localize the effects and likely reduce risks of adverse side effects. However, there are some major concerns that should be considered.

1. A major concern is that the study relies on nociceptive withdrawal reflex assays and to a lesser extent on nocifensive behaviors. Study would be greatly strengthened by using operant based pain assays for at least one of the pain models to increase likelihood of translation to humans.

Operant based assays for local lipid peptides are problematic because of the slow onset of action of these molecules. *In vitro* these peptides require considerable time to partition into the membrane and flip-flop across (PMID: 18044965). Our own observations indicated that the myr-HA peptide required about an hour to traverse CHO cell membranes *in vitro* (data not shown). It would be difficult to pair our peptides with a conditioned stimulus because of the slow onset action. Indeed, operant based pain strategies typically use centrally acting drugs with cannulas for immediate conditioned stimulus pairing. Moreover, as we are observing gender differences, where in established inflammatory pain, female response to the Ap2 inhibitor peptide is delayed, this would further confound the operant based strategy. As an alternative, we have now included non-reflexive dynamic weight bearing assessments in Figure 4L. We see that both free moving male and female mice are able to significantly increase weight bearing on the inflamed paw 48 hours after AP2 peptide treatment but scrambled peptide treated mice do not. We believe that this additional, non-reflexive behavior assessment increases the translatability of our findings.

2. A second major concern is with the data reporting sex difference of AP2 inhibitory peptide treatments, which may be overstated. The difference is not consistently seen, which may reflect the small sample sizes, and when seen is not that impressive, which raises questions of biological relevance. In Fig 4, data are taken from a single experiment with only n=4 mice. Mouse behavior is too variable to draw any conclusion from just 4 mice. Further, the tau calculation mainly points out a difference in the control values between male and female (both decrease but from different starting points). The same comments apply to Fig. 5 for which they used rat data (n=6 for each sex) and appear to infer that the peptide worked better in females than males, although this is not clear in the Discussion. A concern is that there apparently was not any sex dependence of the peptide in mice since only the scrambled controls for mice are shown in suppl fig 6, and in this experiment, the scrambled

controls do not differ between sexes, unlike in rats. Taken together, it seems that there is variability in the assays and when the groups are small (n=4, 6), differences are being observed that really need to be confirmed with larger n's and, importantly, in independent experiments.

We have now increased n values for both mice and rats in Figures 4 and 5. In essence we have repeated the experiment independently and the data continues to hold. If the AP2 inhibitor peptide is given after established pain, females continue to show a delayed analgesic response whereas males recover more rapidly. If the AP2 peptide is given in advance of injury, recovery for females is substantially better than males.

3. Hard to get a grasp on prevalence of AP2 α 2 in DRG from data in fig 1. Panel A looks like a small percentage, while panel B looks like almost all neurons.

The original figure 1B was overexposed, illuminating much of the background. That same figure has been adjusted (equally on both ipsilateral and contralateral sides). This figure better represents the differential distribution of AP2 α 2

4. The AUC data should be represented as individual data points (e.g. as done in Fig 4A) not as traditional bar and plungers (e.g. Fig 4E) to give better appreciation of the spread in each group.

The AUC graphs have been changed to include individual data points.

5. Were there sex differences with the shRNA experiments? If not, this should be addressed, pertinent to point #2.

We did not observe sex differences with the shRNA experiments. This is because recovery cannot be accurately gauged using the spinal nerve injection approach as genetic knockdown requires many days to occur. We have added a point in the Discussion to address this.

6. How explain differences between shRNA and the inhibitory peptide? Eg shRNA affects lifting behavior, mechanical sensitivity, etc, while peptide did not.

It should be noted that shRNA knockdown will knockdown the AP2 α 2 subunit at both peripheral and central terminals, as well as at the cell body, whereas the AP2 inhibitory peptide is only acting at the site of inflammation/injury (i.e. peripheral terminals). Centrally located AP2 α 2 might affect synaptic transmission and since some CGRP nociceptors are polymodal, this might be accounting for the increased mechanical sensitivity observations. We added a paragraph in the Discussion to address this point.

7. The point of testing other peptides in suppl fig 7 is not clear. For example, what was the rationale for selecting these peptides?

We tested other peptides to demonstrate that sequence determines efficacy (Supplemental Fig6A, however duration of action (Supplemental Fig6B) remains conserved. This important because we must use the myr-HA peptide as a proxy to demonstrate lipidated peptide localization to nerve

endings. There are no antibodies that can react to the peptides we designed. We can use these results to demonstrate that lipidated peptides are stable and can be considered a novel method to pharmacologically target nerve endings for disorders such as pain. It will really depend on the sequence of the peptide and what is being targeted.

Minor

1. Need to say what AP2a2 is in the abstract and abstract should clearly state the gender differences if that is kept in the paper.

We have amended the abstract accordingly.

2. Fig 1F, while nice to have representative images and videos, data should be quantified for presentation.

The data was quantified in the original figure. This figure is now Fig1E, and quantification is presented in Fig1F. We have also supplied the representative videos in the journal database.

3. Fig 2 B is not percent of baseline as labeled for the Y axis. It is a ratio, as correctly labeled in parentheses.

This has been corrected and reflected as changes in the y-axis values for appropriate graphs.

Reviewer #2 (Remarks to the Author):

This manuscript reports that AP2a2-mediated endocytosis in CGRP-expressing nociceptors contributes to acute and chronic inflammatory pain in mice and rats. AP2a2 was found to be preferentially expressed in CGRP+ve nociceptors in mouse and human DRG. Knockdown of AP2a2 attenuated PKA-evoked nociceptor excitability and blunted acute nociceptive behavior in mice. Lipidated probes incorporated into membranes of CHO cells and DRG neurons, where they were retained for 72 h. Lipidated peptide-based inhibitors of AP2a2 suppressed acute nociceptive behavior and reversed CFA-evoked inflammatory nociception in mice. They also suppressed surgically-induced nociception in rats. These anti-nociceptive actions lasted for several days. Injection of the lipidated AP2a2 inhibitor led to increased CGRP immunoreactivity in distal layers of the stratum granulosum, suggesting that inhibition of endocytosis suppresses CGRP release in the periphery.

This is an interesting manuscript that builds on the authors' published work showing that AP2 clathrin-mediated endocytosis underlies DRG neuronal sensitization through effects on trafficking of sodium-activated potassium channels. The present study extends this work by examining nociception in animal models. The most striking finding is that lipidated AP2a2 inhibitors have sustained anti-nociception actions in mice and rats. The authors use multiple approaches to support their conclusions. In general, the experiments are clearly described and the results are convincing.

Several issues require attention.

1. Some experimental approaches are not fully described.

a) Fig. 1D shows the effects of AP2a2 shRNA on PKA-induced hyperexcitability of DRG neurons. How was PKA activated in these experiments?

Thank you for bringing this to our attention. This was inadvertently excluded. DRG neurons were treated with the cell-permeant PKA stimulator Sp-cAMPs (cAMP analog). We have now included this in the methods.

b) Fig. 2 shows the effects of AP2a2 shRNA given pre- or post-inflammation. Please clarify the timing of shRNA administration relative to the CFA injection since this is not clearly stated in the methods.

Time of injection are labeled on the x-axis, however, this has been further clarified in the Results section.

2. Some results require more complete quantitation.

a) Fig. 1A shows preferential localization of AP2a2 in CGRP+ve DRG neurons. The results would be more convincing if the authors quantify the proportions of CGRP+ve and IB4+ve neurons that express AP2a2 in several animals?

We have characterized AP2 α 2/CGRP co-localization in several animals. We always noted a differential distribution of AP2 α 2 as clearly depicted in Figures 1A&B. We have not observed AP2 α 2 in any IB⁺ neurons. Indeed, for the types of experiments the Reviewer is suggesting to be meaningful, the experiment should be performed with multiple DRG markers (TrpV1, Substance P, Neuropeptide Y, Mrgprd included) but we feel this is beyond the scope of the manuscript. These are however important experiments we plan to conduct in the future, to understand exactly the distribution of AP2 α 2 in the DRG. Nonetheless, we would like to highlight that this differential distribution of AP2 α 2 in CGRP neurons is consistent with the behavioral data we observed: very strong thermal responsiveness but mostly modest to no mechanical responsiveness. These are similar to previous CGRP ablation experiments showing a predominant thermal deficit effect. (PMID: 23523592).

b) Fig. 1D shows representative traces of the effects of AP2a2 shRNA on PKA-induced hyperexcitability of DRG neurons. Pooled data of action potential firing should be shown and analyzed statistically.

We have previously extensively characterized the effect of AP firing in DRG neurons during AP2 inhibition (PMID: 28982974). In addition to the Western analysis, immunohistochemistry, this data was included to simply further demonstrate that AP2 α 2-dependent signaling is compromised after shRNA knockdown.

c) Can the authors quantify the effects of AP2a2 inhibitors on CGRP concentrations measured by ELISA rather than rely on semi-quantitative IF?

Our semi-quantitative approach was to simply demonstrate that CGRP basal signaling is disturbed by AP2 peptide inhibitor treatment. The consequences of this data reinforces the observed behavioral data but also challenges the current dogma on spatial localization of CGRP

nociceptor terminals in the epidermis (PMID: 15629699). The literature suggests that CGRP nociceptor terminals prematurely terminate compared to Mrgprd⁺ neurons which innervate the very superficial layers of the skins. Our IF data actually demonstrates that CGRP nociceptor terminals proceed much farther into the epidermis. Quantitative ELISA on isolated hindpaws or from cultured neurons would require extensive analyses. We would need to look at both basal, and stimulated (potassium chloride or capsaicin) conditions. We believe that these types of experiments are beyond the scope of the manuscript.

3. The authors examined the distribution of lipidated probes by localizing a lipidated HA peptide. They extrapolated these findings to the distribution of lipidated AP2a2 inhibitors. However, this may not be appropriate since the AP2a2 inhibitor may preferentially target AP2a2 in CGRP⁺ neurons. Can experiments be designed to localize the lipidated AP2a2 inhibitors?

We agree that it would have been better to examine directly the localization of the AP2 inhibitor peptide. Beyond immunohistochemistry there would be no way to be able to do this to gain the resolution necessary to distinguish between neuronal and non-neuronal tissue. That is why we included Supplemental Figure 6. We show that various peptides can offer different efficacies but similar durations of action. The HA peptide immunohistochemistry therefore acts as a good proxy for small lipidated peptide localization to nerve endings.

4. Fig. 3 shows uptake of lipidated HA probes into afferent fibers. What is the evidence that these are afferent nerve fibers? Is it possible to co-label tissues with a neuronal marker? Do the authors propose preferential probe uptake into neurons and, if so, how does this occur?

We agree that this experiment needed to be conducted. We have now included a panel (Fig. 3C) showing that the HA-peptide penetrates neuronal endings using the cytoplasmic pan-neuronal marker PGP9.5. Myristoylated peptides will penetrate any lipid accessible compartment, that is why we see labeling in other cell types as well.

5. Supplemental Fig. 3 claims to show retention of lipidated HA probes in the plasma membrane of CHO cells and DRG neurons for several days. However, the images are low magnification/resolution and the entire cell is labeled. Higher power confocal images are required to demonstrate that the probe is confined to the plasma membrane. Given the rapid turnover of membranes, does the probe traffic to and from the plasma membrane and endosomes? Panel A lacks a scale bar. A control for selectivity of the HA antibody is required.

We have added an additional high magnification image of CHO cells incubated with the HA-peptide (collected 6 hours post exposure) as panel B. As can be seen, the HA-peptide localizes to the membranes of cells. We are confident that this image strengthens our position that the HA-peptide primarily localizes to the membrane. In regards to potential trafficking of the peptide: it is predicted to associate with the cell membrane and membrane bound compartments. During an endocytotic event, the HA-peptide would be expected to traffic along with the early endosome. However, during pinocytosis, membrane recycling should occur and thus we are observing longevity of peptide within CHO cells and DRG neuronal membranes (at least 72 hours). Lipid turnover itself is a very slow process (PMID: 11264283 (ref 55)). This experiment would be best done using live cell imaging and a fluorescent labeled peptide, but is beyond the scope of this

manuscript. Scale bars and control images have been added, however, due to space, the 6-hour time point (Supplemental Fig. 3A) has been removed and replaced with the high magnification image.

6. The authors propose that defective CGRP release from the peripheral endings of nociceptors might explain the anti-nociceptive properties of AP2a2 knockdown and inhibition. This conclusion is based on increased CGRP staining in peripheral tissues in animals treated with the lipidated AP2a2 inhibitor. Given that CGRP along with Substance P (SP) is a major mediator of neurogenic inflammation in the periphery, how to the authors explain that the AP2a2 inhibitors do not affect paw inflammation?

We administer the AP2 peptide 24 hours after CFA, which is 24 hours after full-blown inflammation. We would not expect the AP2 peptide inhibitor to accelerate the resolution of the inflammation. In pre-emptive AP2A2 knockdown, while we note a significant reduction in pain behavior, inflammation was not affected. This could be due to the fact that we are not achieving full AP2 α 2 knockdown (i.e. we observed a ~60% reduction). The remaining AP2 α 2 may still allow for sufficient peptide release for paw inflammation to still occur.

Are SP levels similarly affected?

We did not examine SP. But since Substance P is often co-localized with CGRP containing LDCVs, we surmise that SP would also be affected.

The central release of CGRP in the spinal cord mediates pain transmission. Are there changes in CGRP levels in the central projections of nociceptors in animals treated with AP2a2 inhibitor?

We did not look at CGRP in central projections after AP2 peptide inhibitor. Because Ap2 inhibition affects nociceptor excitability, we might expect that there is CGRP retention within the central terminals (and glutamate too). However, lipidated peptides will not cross the blood brain barrier (PMID: 21053136), so any effects we see would be the indirect effect of local AP2 α 2 inhibition at peripheral terminals.

7. In the discussion, the authors also raise the possibility that altered trafficking of ion channels might also explain the anti-nociceptive actions of AP2a2 inhibitors. This is based on their past work. There are several reports that inhibitors of clathrin and dynamin can inhibit nociceptor hyperexcitability and nociception in rodent models. These effects have been attributed to inhibition of endosomal signaling of GPCRs – CLR, NK1R, PAR2. The authors may wish to speculate that AP2a2 inhibitors could also block endosomal signaling of GPCRs that mediate pain transmission.

As we have shown in Fig 1D and have previously shown (PMID: 20962237, PMID: 28982974), cell-permeating direct activators of PKA endocytose K_{Na} channels. The net effect is a loss of firing accommodation, the signature firing pattern of nociceptor sensitization. K_{Na} channel endocytosis is clathrin-dependent as we also showed that inhibiting AP2 prevented clathrin recruitment to K_{Na} channels after PKA stimulation (PMID: 28982974). In terms of excitability, nociceptor endocytosis of potassium channels would supersede endocytosis of GPCRs. Furthermore, nociceptors are endowed with both pro-nociceptive and anti-nociceptive GPCRs.

It's hard to know what the net effect of blocking GPCR endocytosis is in our model. However, one study showed that in beta2 arrestin deficient mice, there is an observed increase in inflammatory pain behaviors (PMID: 27538456). So, at this time, we are not comfortable speculating on the role of GPCR endocytosis is in this process.

8. The studies of human nociceptors add translational relevance but are limited because they are of nociceptors from a single donor. Ideally, observations would be replicated in DRG from several subjects. The proportion of different types of neurons expressing AP2α2 could then be quantified.

We agree that assessment from multiple donors would have added to the study. It would have been informative to know what the AP2α2 levels are in chronic pain subjects. Unfortunately, the cost of getting those specimens from Anabios would have been prohibitive. For us to conduct those types of studies from human donors here at our institution would be time prohibitive.

Reviewer #3 (Remarks to the Author):

This manuscript entitled “Inhibiting endocytosis in CGRP+ nociceptors as a treatment for inflammatory pain” by Powell et al. examine local disruption of nociceptor endocytosis and use various inflammatory pain models to characterize the in vivo contribution of extra-synaptic AP2 clathrin-mediated endocytosis (AP2-CME) to inflammatory pain.

Overall, the data presented do not provide further evidence for “peptidergic nociceptors as executive regulators of inflammatory pain” or support “nociceptor endocytosis as a promising target for local, specific, and long-lasting treatment of inflammatory pain.” The experiments described here are not truly validating the proposed target in inflammatory pain. The in vivo data supporting the claims in the manuscript are problematic:

Figure 1.

(1)Panel C (right): Is the blue bar that is labelled as Control, the scrambled shRNA or a different Control?

The blue bar is scrambled shRNA. This has been clarified.

(2)Panel D: Data for PKA stimulation under control conditions is missing, to enable interpreting the data with the scrambled and AP2 shRNA in perspective.

PKA-induced hyperexcitability of DRG neurons has been shown by us and other groups (PMID: 16120663; PMID: 17021029; PMID: 20962237; PMID: 28982974). The only comparisons that can be made is scrambled vs AP2 shRNA where the nerve injection must be accounted for.

(3)Panel E: A baseline 5% formalin response cohort is not shown, making it hard to appropriately compare the results with either the scrambled or AP2α2 shRNA groups.

Addition of this group would not be appropriate for the formalin assay, as the nerve injection serves as an additional variable to the experiment. A naïve control group would not function as a proper control under these conditions. There would be two variables in the naïve group,

prohibiting direct comparisons and would not justify the use of extra animals. For these reasons, we maintain that the scrambled shRNA groups functions as the proper control group in these experiments.

(4)The number of licks, lifts and flinches following 5% formalin are quite high with the scrambled peptide for an N=6 and is hard to put into perspective without having a control group to understand the true baseline. How exactly were the behaviors scored from the video? Are these male or female mice, or is this pooled data? This is not clarified in the legend.

The Reviewer might have missed this. The scoring parameters were outlined in the Methods section: “All behaviors were scored for a full minute, every five minutes, for 90 minutes of video recording. Scorers were blinded to experimental conditions.” This data is pooled from males and females. This was clarified in the legend.

Figure 2.

(1)Absence of baseline CFA response in animals to compare with the scrambled (n = 8) and AP2α2 (n = 8) shRNA groups. Are these pooled male and female?

Baseline behaviors for each group are included to the left of each y-axis. These data are pooled males and females. This has been clarified in the legend.

(2)Negligible difference in thermal sensitivity, in the AP2α2 group (less than 3 sec) and the AP2α2 is still significantly sensitive to thermal stimuli at day 8.

We respectfully disagree. Although the magnitude of the difference between the 2 groups is 2 seconds, 24 hours following CFA injection (when nociception is maximal) pre-emptive knockdown of AP2α2 was sufficient in significantly decreasing thermal sensitivity. 2 seconds here is profound. This does not render the effect negligible as the data was statistically significant different using the most stringent statistical testing conditions.

(3)The mechanical sensitivity change is also not particularly impressive. Data are better plotted as withdrawal threshold instead of mean percent of baseline to understand a true response.

The lack of a strong effect on mechanical nociception falls in-line with our hypothesis, and findings, that inhibition of endocytosis would produce a more profound effect in thermally responsive nociceptors compared to mechanosensitive nociceptors (PMID: 23523592).

(4) A 5 sec withdrawal latency in the scrambled shRNA group and a 2 second change in the AP2α2 shRNA makes it hard to interpret the results.

We respectfully disagree. Post-CFA shRNA mediated knockdown of AP2α2 produces a statistically significant decrease in thermal sensitivity 6 days following shRNA injection under stringent statistical testing conditions.

Figure 4.

(1) Panel A: A control group is missing to enable appropriate comparison to either the scrambled or AP2α2 shRNA groups. It is odd that licking behavior shows a difference between the scrambled and AP2α2 groups but lifting and flinching do not show a difference. Are these pooled male and female rats?

As previously stated above, the scrambled shRNA group functions as an appropriate control for this assay. However, in this experiment, we have injected our peptidomimetic directly into the ipsilateral hind paw of the animals. Nevertheless, naïve mice would not aid in interpretation of results because of additional variables. As for the behavior difference in the measured pain modalities (licking, lifting, and flinching) compared to Figure 1, this experiment utilized a local injection of the peptidomimetic. At the sight of injection, formalin would cause extensive tissue damage and nerve ending fixation (to a degree), so the number of intact peripheral fibers that the peptidomimetic infiltrated would be lower, thus dampening the observed behavioral effect. The Reviewer seems to have missed this, but we had stated this in the Results (with citation # 43). These data are pooled males and females. This has been clarified in the figure legends.

(2) A second difference in data presented in panels C and D with an N=4 is hard to justify as an analgesic response. Thermal sensitivity in the CFA model is generally variable across time. Hence, it is difficult to understand how such minimal variability was achieved with an N=4 in the experiments presented. Given the minimal response observed in data presented in panels C and D, it is not clear whether the additional analysis presented in panels E-K are truly valid or provide an over interpretation of the response to the AP2shRNA. Panel K shows a difference between the two groups only on Day 2.

As per Reviewer 1, we have increased the number of animals to not only decrease variability but conduct an additional independent experiment. With the addition of the second set of experiments, statistical significance was maintained. With this additional data and analysis offered in Fig. 4E – K, the overall conclusions of our manuscript have not changed. The difference in panel K further illustrates the specificity of the effect of inhibiting endocytosis in predominantly thermal responsive peripheral nerve afferents.

(3) In both Figures 2 and 4 it is not clear as to whether the thermal sensitivity was measured at the same time on each day.

All animal experimentation was conducted during the light phase each day. We have clarified this in the methods.

Figure 5.

(1) Panel D- it is not clear that the response in the ipsilateral AP2α2 group has a late phase and an early phase. Is the statistical comparison to the sham ipsilateral group? Again, the data here is hard to interpret in the absence of a control group with no surgery.

The data in panel D displays an apparent early (day 1 – day 2) and late (day 4 – day 6) phase that is present in the control condition. The statistical comparison is made to the scrambled peptide ipsilateral group. A ‘no surgery’ control group would serve to be rendered redundant in this experiment due to the presence of withdrawal latencies from the uninjured contralateral paw.

(2) In these experiments, the treatment paradigm of injecting peptide into the muscle then suturing and injecting more peptide after suturing is a bit odd.

The Reviewer may have overlooked this. We originally stated in the Results section “For this assay, we simulated a potential clinical application schedule for the AP2 inhibitory peptide; sub-cutaneous administration into the hind paws of rats 6 hours before incision, and then a series of smaller sub-cutaneous and intra-muscular injections immediately following incision”. To elaborate, the injection schedule was structured in this sense to give a higher degree of translatability to a clinical real-world scenario where a patient would receive a dose before surgery, and then a subsequent dose immediately following the procedure.

Supplementary Figure 7.

(1) Panel A: it is hard to interpret the effects with P1-P4 in the formalin model. The AP2 α 2 group (n=6) shows no decrease in the lifting behavior compared to the scrambled peptide, but the P1-P4 groups (n=3) reduce lifting behavior, but not licking behavior or flinching behavior.

(Supplementary Fig. 7 has been changed to Supplementary Fig. 6) Each peptide (P1 – P4) corresponds to a different peptide sequence. From this data we conclude that varying the sequence of the AP2 peptidomimetic alters the apparent efficacy of each peptide in the formalin assay potentially reflecting variation in affinity for the AP2 complex *in vivo*.

(2) It is odd that the response of the scrambled group and the AP2 α 2 group are remarkably similar to that in Figure 1.

The data for the ‘AP2 α 2 peptide’ and ‘scrambled peptide’ groups represented in Supplemental figure 7A (Now Supplemental figure 6A) are taken from Figure 1E and superimposed onto the same axis as P1-P4. This is now stated in the figure legend.

(3) In Panel B, are the responses to ipsilateral Ch1001 and Ch1002 being compared to the contralateral Ch1001 and Ch1002, or to the scrambled or the AP2 α 2 group?

The ipsilateral responses for the ‘Ch1001’ and ‘Ch1002’ groups are being compared to the ‘scrambled peptide ipsilateral’ group.

Overall, it is not clear how the proposed mechanism supports the therapeutic possibility, given the negligible effects observed overall in the 3 pain models. It does not seem that the experiments described here are truly validating the proposed hypothesis and target.

We respectfully disagree. The effects were profound, backed by stringent statistical analyses.

Reviewers' comments:

Reviewer #1 (Remarks to the Author):

The authors have tried to address some of the concerns, but major concerns remain unresolved. Some of these are:

1. Biological significance of the changes seen, overall and especially the putative sex differences. We are still looking at about 2-3 sec differences, which while this reaches statistical significance, remains a concern. It was hoped that by increasing the number of samples that this would be strengthened, but unfortunately, the differences are still very small. This was also brought up by other reviewers.

2. The explanation for the sex differences is still not very clear. Again, the differences are small. The fact that data are shown as pooled male and female really speaks to the question of relevance of any slight sex difference.

2. Inclusion of weight bearing assay is good, but puzzling since the main phenotype is claimed to be thermal, so why do a mechanical test? Indeed, there is only a slight difference observed in this assay, which while it reaches statistical significance, is not convincing. In addition, not all operant assays require conditioned place preference/aversion as stated in the response.

Reviewer #2 (Remarks to the Author):

Thank you for responding to my suggestions for revision.

The only remaining concern is the study of DRG neurons from a single human donor.

Reviewer #1 (Remarks to the Author):

The authors have tried to address some of the concerns, but major concerns remain unresolved. Some of these are:

1. Biological significance of the changes seen, overall and especially the putative sex differences. We are still looking at about 2-3 sec differences, which while this reaches statistical significance, remains a concern. It was hoped that by increasing the number of samples that this would be strengthened, but unfortunately, the differences are still very small. This was also brought up by other reviewers.

In our manuscript, there are multiple principal findings: AP2 α 2 is expressed in subset of a subset of DRG neurons, nociceptor endocytosis is required for both the development and maintenance of inflammatory pain and lipidated peptides provide effective and long-lasting reductions in pain behavior when locally administered. We did not set out to examine gender differences in pain. This study was supported by the NIH, and the NIH mandates that experimentation be performed on both sexes, that data can be pooled if there are no differences, but should be segregated if there are. However, big or small the gender difference may be, if the difference is statistically significant, the principles of statistics demand that data be segregated. When it came to how male and female mice responded to our AP2 inhibitor, we noted substantial and significant differences in responses and pain recovery curves. This is why we are obliged as per NIH guidelines to report on these differences in this manuscript. The Reviewer rightfully pointed out that in our first submission, despite data already achieving statistical significance, that we should increase *n* values to be certain that the sex differences hold. Unhesitatingly we did this, increased *n* values and showed that females and males respond differently to the peptide if pain is already established, or when pre-emptively delivered before injury-induced inflammation. We have now modified Figures 4 and 5 and the Supplemental figures to make this simpler and clearer in the manuscript. Nonetheless, the 2-3 second point the reviewer brings up, which is in reference to what Reviewer 3 initially brought up pertains to the genetic knockdown data (Fig 2A). We did not observe sex differences during genetic knockdown, therefore the data was pooled. In our rebuttal, we pointed out that genetic knockdown resulted in 60% knockdown of the AP2 α 2 subunit (Fig 1C). That means that there was still about 40% AP2 α 2 subunit still available. In this AP2A2 knockdown approach included was a surgery, a 7-day recovery period, and an incomplete genetic knockdown. These may have been contributing factors to the lack of gender difference. Nonetheless, in our hyperalgesia paradigm, we observe baseline withdrawal latencies to be at 10 sec, with peak CFA hyperalgesia 24 hours after, at withdrawal latencies of 2 sec, making the paw withdrawal latency difference of 8 sec under peak CFA inflammation. With 60% Ap2A2 knockdown, we see a statistically significant 3s reduction in paw withdrawal latency at this peak point. That's a 38% reduction of peak pain behavior, which improves to 50% pain reduction at 48 and at 72 hours after the CFA injection. To put this in perspective, current analgesics provide about 30% pain relief for patients in chronic pain (PMID: 26844640). Again, we are not achieving complete AP2 α 2 genetic knockdown but obtaining significant and substantial effects.

2. The explanation for the sex differences is still not very clear. Again, the differences are small. The fact that data are shown as pooled male and female really speaks to the question of relevance of any slight sex difference.

Again, the explanation of the sex difference in response to the Ap2 inhibitor peptide has been altered in hopes to help make it clearer. Figures 4 and 5 have been streamlined and excess graphics have been moved to the Supplement in order to aid in effectively communicating the data. With that being said, we cannot but help but feel as though the reviewer might be confused with the presentation of some of the data and the conclusions drawn from them. As stated above, the AP2 α 2 shRNA mediated knockdown data (Fig 2) was pooled due to a lack of gender difference. The AP2 inhibitor peptide, our pharmacological approach, however, showed sex-dependent differences. The pharmacological data in the previous version of the manuscript presented both pooled, and separated male and female data for absolute paw withdraw latencies, area under the curve, and recovery kinetics. In the revised manuscript, in the principal figures, consolidated male and female data is presented and the gender segregation has now been moved to the Supplement. However, to address just how relevant our data is, for males, the one-time local injection of peptide reduced CFA-induced paw withdrawal latency by 4 sec, which represents a 50% reduction in pain behavior, 24-hours after a single dose administration (new Figure 4E and Supplemental Figure 4C). Contrast this with females: in females, 24 hours after a single injection, Ap2 peptide inhibitor peptide and scrambled peptide have near identical paw withdrawal latencies. The data presented in the new Figure 4F statistically compares Ap2 inhibitor males vs Ap2 inhibitor females. Paw withdrawal latencies were significantly different. We hope this is now clearer. At 48 hours post peptide injection is where the Ap2 inhibitor in females start to display a significant reduction in pain behavior compared to scrambled peptide treated females (Supplemental Fig 4H). There is a delay in effect for females: this is a sex difference and the difference is huge.

3. Inclusion of weight bearing assay is good, but puzzling since the main phenotype is claimed to be thermal, so why do a mechanical test? Indeed, there is only a slight difference observed in this assay, which while it reaches statistical significance, is not convincing. In addition, not all operant assays require conditioned place preference/aversion as stated in the response.

The inclusion of the dynamic weight bearing data helps to compliment the other non-evoked assay employed in this manuscript (formalin assay) as well as present another measure of ongoing inflammatory pain behavior (PMID: 24888508). We respectfully disagree with the reviewer that this assay functions as a test for mechanosensation. The dynamic weight bearing assay indirectly measures non-evoked inflammatory pain behaviors by measuring the amount of weight borne by an individual limb. This would presumably be a measure of coping behavior (the willingness of the animal to utilize the limb/place weight on the limb, indicates a decrease in the need for the animal to cope, i.e., redistribute weight). This distinctive pain behavior is independent of exteroception (sensitivity to external threats) and more closely resembles ongoing internal pain, mediated by TrpV1-positive nociceptors (PMID: 30532001). Human corollaries exist in

diabetic patients with Charcot foot (PMID: 23705057) where patients describe a dull pain when walking, but lack sensation of acute suprathreshold mechanical sensation (mediated by A δ fibers) suggesting that ongoing pain is mediated by C-fibers. A previous study conducted by the Porreca group utilized weight bearing to characterize PKR1 knockout mice (PMID: 16793879). The PKR1 receptor is expressed in TrpV1 neurons and in PKR1 knockout mice, these mice failed to exhibit CFA-induced thermal hyperalgesia and failed to exhibit CFA-induced weight bearing but these mice retained CFA-induced tactile allodynia. Like our data, the Porreca group showed CFA-induced weight bearing more closely followed thermal sensitivity, not mechanical sensitivity. The von Frey hair provides a pinpoint, acute stimulation, not allowing for sensory adaptation and thus cannot be compared to how weight is distributed over the entire surface of the paw. With respect to magnitude, it cannot be overlooked that most weight bearing studies use static weight bearing, whereas we used dynamic weight bearing. In static weight bearing the difference is that the animals are restrained and weight is measured only on the hind paws, so the effect is more easily distinguishable due to an inability to distribute extra weight across the forepaws. In the dynamic weight bearing apparatus, the animals are allowed to freely roam the enclosure while a floor sensor records weight across all paws, a more naturalistic examination of pain behavior. Thus, the animal is granted the opportunity to distribute additional weight across the forepaws as well as the contralateral hind paw. Alongside this, in order to capture analgesia experience by female animals in the cohort (because again there is a delay in effect), we decided on taking the 48-hour time point for our weight bearing analysis, which adds an additional confound, natural recovery, that might have further dampened the observed effect. Nevertheless, we still observed a statistically significant increase in weight borne by the ipsilateral paw in the AP2 peptide group that is not present in the scrambled group 48 hours after a single dose administration.

The Reviewer requested non reflexive behavior to bolster clinical relevance of our findings. We did that. Dynamic weight bearing is a measure of non-reflexive inflammatory hyperalgesia (PMID: 25738619; PMID: 31746599). The reviewer suggested operant based assays, but the problem as we see it with operant conditioning for a local analgesic, is that we must directly apply via needle, drug into the painful, inflamed paw without the use of general anesthesia. We cannot conceive how this procedure can lead to successful direct operant conditioning types of experiments, especially since the peptide has a slow onset of action. Even when lidocaine is used for operant conditioning on CFA treated animals for example, it's not given at the site of the inflamed paw but further up at the popliteal fossa to induce nerve block (PMID: 21219650) or intrathecally (PMID: 22609247). The mode of action of our peptide does not work like this. Our data showed increased weight bearing, (again a measure of ongoing pain) after peptide injection and the data was statistically significant and we believe this raises the relevance of our findings.

Reviewer #2 (Remarks to the Author):

Thank you for responding to my suggestions for revision.

The only remaining concern is the study of DRG neurons from a single human donor.

Thank you for your suggestions for revision, it improved our manuscript. Due to difficulty in sourcing primary donor tissue from a distributor, we have now explicitly stated that the human data was obtained from a single donor in the Abstract as well as the Results section.

REVIEWER COMMENTS

Reviewer #4 (Remarks to the Author):

The authors have developed a method for targeting AP2alpha2 (AP2A2) to inhibit pain. They show that knockdown of AP2A2 or a peptide that targets AP2A2 inhibits inflammatory and post-surgical pain and that this approach seems to promote CGRP retention in nociceptors in the skin suggesting that these neurons are selectively inhibited by the AP2A2 targeting approach. Overall, this is an exciting study suggesting a new approach for peripheral pain treatment. The results are mostly convincing, but there are some areas where the paper could be improved.

1) I think that the abstract overstates the selectivity of the approach. The experiments suggest that the CGRP population is selectively targeted, but AP2A2 is broadly expressed at the mRNA level so other cellular targets cannot really be excluded. I also think that the sex differences are stated incorrectly. There do not seem to be mechanistic differences shown in this study, instead there are sex differences in therapeutic response. Those are not exactly the same thing.

2) The authors are inconsistent on how they describe CGRP nociceptors. In the intro they are responsible for thermal and mechanical, yet in the results it is thermal and chemical. In mice, there is pretty strong evidence that CGRP+ nociceptors are responsible for thermal and chemical nociception but likely not mechanical.

3) The mousebrain.org single cell dataset shows pretty clearly that AP2A2 gene is expressed ubiquitously in DRG neurons. Maybe it is not translated in a subset of neurons, which can be the case as shown for other mRNAs like NFH, but the mechanical effect in CFA could also be because the IHC just does not reveal expression in some IB4+ neurons in the mouse.

4) I really struggle to understand how the authors have assessed sex differences. They show some differences in the kinetics of the drug effect between males and females, but this can simply be a pharmacokinetic issue. Moreover, as far as I can tell, the authors have not done any three-way anovas that specifically test for sex differences in responses. Rather, they show some differences in drug response using a t-test at what seem to be specific time points. I think there likely is some sex difference here, but it is small, and the authors are emphasizing these differences based on some rather selective views of the data.

5) Dose response studies for the peptide are needed at some point.

We thank the Reviewer for insightful comments. Acting upon the criticisms, we feel our resubmitted manuscript is very much improved. Below are our responses to the Reviewer's comments.

The authors have developed a method for targeting AP2alpha2 (AP2A2) to inhibit pain. They show that knockdown of AP2A2 or a peptide that targets AP2A2 inhibits inflammatory and post-surgical pain and that this approach seems to promote CGRP retention in nociceptors in the skin suggesting that these neurons are selectively inhibited by the AP2A2 targeting approach. Overall, this is an exciting study suggesting a new approach for peripheral pain treatment. The results are mostly convincing, but there are some areas where the paper could be improved.

1) *I think that the abstract overstates the selectivity of the approach. The experiments suggest that the CGRP population is selectively targeted, but AP2A2 is broadly expressed at the mRNA level so other cellular targets cannot really be excluded. I also think that the sex differences are stated incorrectly. There do not seem to be mechanistic differences shown in this study, instead there are sex differences in therapeutic response. Those are not exactly the same thing.*

The original characterization of the two distinct alpha-adaptin isoforms by Ball et al (PMID: 7593326) used an immunohistochemical approach to describe the expression pattern of each isoform in the spinal cord. Of note, expression of the $\alpha 2$ -isoform was constrained to the superficial laminae of the spinal cord, whereas the $\alpha 1$ -isoform had ubiquitous expression throughout the entirety of the spinal cord. This expression pattern suggested that there is differential expression of the AP2A2 within peptidergic nociceptors as it is well known that the superficial dorsal horn are the sites of termination for incoming primary CGRP afferents (PMID: 21768598). Additionally, the selectivity of our observed effects, which was precipitated by genetically knocking down AP2A2, supports the assertion that AP2A2 is preferentially expressed in peptidergic neurons. As is stated below in comment #3, the Reviewer is claiming broad AP2A2 DRG expression based on mousebrain.org, which used scRNAseq data to assemble its' database. The mousebrain.org database is a powerful tool for understanding the expression profiles of various genes through the mouse nervous system. However, it must be caveated that the database was constructed based on two endeavors using scRNAseq experiments in rodents (PMID: 30096314 and PMID: 25420068). We believe that the apparent ubiquitous expression of AP2A2 in DRG neurons as reported by mousebrain.org is likely an artifact arising from the analysis paradigms employed in PMID: 30096314. In this paper the authors formed subgroups of neurons using general markers (i.e., tyrosine hydroxylase (TH) for non-peptidergic neurons and CALCA for peptidergic neurons) derived from PMID: 25420068. In each instance, a fixed number of cells were analyzed to make these subdivisions, then within each subgroup, further subgroups were created based on differences in RNA expression between each sub-subgroup (i.e., within the TH subgroup, the NP1.1 and NP1.2 sub-subgroups were determined based on differences in RNA expression between TH⁺ neurons). The subdivision of neuronal subtypes from a parent group artificially inflated apparent AP2A2 expression in subsequent subgroups, due to decreasing numbers of neurons belonging to each sub-subgroup, exaggerating the abundance of AP2A2 RNA (TH

subgroup; 282 cells, AP2A2 abundance = 0.4929 vs NP1.1 sub-subgroup; 72 cells, AP2A2 abundance = 1.7778). Importantly, the TH subgroup has relatively low AP2A2 expression (TH subgroup; AP2A2 abundance = 0.42929) when compared to the peptidergic neuron subgroup (PEP1.1 subgroup; 117 cells, AP2A2 abundance = 2.0513). Moreover, the TH⁺ sub-subgroups derived from the TH subgroup that had higher abundances of AP2A2, also had high levels of neuropeptides: CALCA (NP2.2 sub-subgroup; 88 cells, CALCA abundance score = 2068.2, AP2A2 abundance score = 2.9659), CALCB (NP2.1 sub-subgroup; 45 cells, CALCB abundance score = 2092.8, AP2A2 abundance score = 3.1333), and BNP (NP3 sub-subgroup; 132 cells, NP3 abundance score = 9045.2, AP2A2 abundance score = 3.7045). (Classifications, cell numbers, and gene abundance were taken directly from mousebrain.org; a database compiled with data from PMID: 30096314 and PMID: 25420068). Therefore, when carefully analyzing the subgroups of neurons in the original papers, mousebrain.org does not suggest ubiquitous expression of AP2A2 in the DRG, but in fact, this database supports our assertion that AP2A2 is preferentially expressed in peptidergic neurons. In other words, due to misclassification in mousebrain.org, populations of peptidergic neurons were incorrectly grouped together with non-peptidergic neurons thereby decreasing the apparent abundance of AP2A2 in peptidergic neuronal populations while simultaneously increasing the apparent abundance of AP2A2 in non-peptidergic neuronal populations. Thus, we believe our abstract is accurately reporting the findings on a preferential localization of AP2A2 to CGRP⁺ neurons based on prior findings (PMID: 7593326), our own IHC and behavioral data and indeed based on proper categorization of neurons in the mousebrain.org database. With respect to the gender differences, we state “We evidenced sexually dimorphic recovery responses to this pharmacological approach” which we believe is synonymous to ‘...therapeutic response’. We do not state there are mechanistic gender differences in inflammatory pain within the Abstract, but speculate on it in the Discussion.

2) *The authors are inconsistent on how they describe CGRP nociceptors. In the intro they are responsible for thermal and mechanical, yet in the results it is thermal and chemical. In mice, there is pretty strong evidence that CGRP+ nociceptors are responsible for thermal and chemical nociception but likely not mechanical.* We have systematically changed the wording in our manuscript from “mechanical” to now refer to as Von Frey sensitivity or thresholds. Some TrpV1/CGRP nociceptors can obtain mechanical sensitivity during inflammation (PMID: 29241539). These are the silent nociceptors (PMID: 20948530). These nociceptors do not innervate the epidermis and therefore are not as responsive to Von Frey. But as reported by our dynamic weight bearing, pharmacological local inhibition of did significantly increase weight bearing, and we now discuss that this is possibly due to action on these silent nociceptors. Prior work by the Stucky lab showed CGRP modulation of Von Frey sensitivity (PMID: 29925650, ref 7) Nonetheless, the Von Frey effects we observed were small and we conclude likely due to the indirect effects of CGRP.

3) *The mousebrain.org single cell dataset shows pretty clearly that AP2A2 gene is expressed ubiquitously in DRG neurons. Maybe it is not translated in a subset of neurons, which can be the case as shown for other mRNAs like NFH, but the*

mechanical effect in CFA could also be because the IHC just does not reveal expression in some IB4+ neurons in the mouse.

We describe the issues of the data derived from mousebrain.org above in point #1. Again, the Von Frey effect was small and transient in CFA could have been the result of decreased CGRP release in the spinal cord, resulting in decreased central sensitization. Our general observations are that knocking down AP2A2 and locally inhibiting endocytosis affected specifically thermal hyperalgesia reinforcing the notion that APA2 is preferentially expressed in CGRP⁺ nociceptors.

4) I really struggle to understand how the authors have assessed sex differences. They show some differences in the kinetics of the drug effect between males and females, but this can simply be a pharmacokinetic issue. Moreover, as far as I can tell, the authors have not done any three-way anovas that specifically test for sex differences in responses. Rather, they show some differences in drug response using a t-test at what seem to be specific time points. I think there likely is some sex difference here, but it is small, and the authors are emphasizing these differences based on some rather selective views of the data.

We agree with the reviewer that our statistical approach to analyze the sex difference to peptide response was erroneous and gives the appearance that we are selecting data points. As suggested by the Reviewer, we now conduct three-way ANOVA to specifically test for sex dependent responses to local inhibition of nociceptor endocytosis. Using three-way ANOVA the data still holds. We observe the same time-dependent significant differences in responses to the peptide in males vs. females in the two different pain models we studied. The sex-dependent differences we observed cannot be ascribed to sex differences in pharmacokinetics, because the peptide was applied directly to the site of action, circumventing absorption, distribution, metabolism, excretion issues. There is an inherent sex difference in response to local nociceptor endocytosis inhibition.

5) Dose response studies for the peptide are needed at some point.

We agree that should the peptide be developed into a drug (and one day we hope it does), dose-response studies would be absolutely necessary. To conduct these studies would require an immense number of male and female rodents for each dose of peptide, respective controls, pain model and type of behavioral assay. It would also require further justification for IACUC approval. We feel that these types of studies are well beyond the scope of the current study.

REVIEWERS' COMMENTS

Reviewer #4 (Remarks to the Author):

The authors have addressed my comments in a satisfactory fashion. The new analysis of the sex differences improves the paper.

I have 2 notes to pass along to the authors/production office.

1) Von Frey should be von Frey. It is misspelled throughout.

2) I understand the authors' argument about the mousebrain.org data (somewhat) but the Usoskin paper and the Hockley et al colonic projecting sensory neuron papers show the same result, AP2A2 gene expression in DRG neurons is ubiquitous. The Ginty lab paper (Sharma et al.) also shows that AP2A2 is nearly ubiquitous, although it is excluded from one population. I suppose it is possible that the mRNA is only translated in a population of nociceptors, and these are the CGRP-positive nociceptors.

The authors have addressed my comments in a satisfactory fashion. The new analysis of the sex differences improves the paper.

I have 2 notes to pass along to the authors/production office.

1) Von Frey should be von Frey. It is misspelled throughout.

Thank you bringing this to our attention. This has been corrected.

2) I understand the authors' argument about the mousebrain.org data (somewhat) but the Usoskin paper and the Hockley et al colonic projecting sensory neuron papers show the same result, AP2A2 gene expression in DRG neurons is ubiquitous. The Ginty lab paper (Sharma et al.) also shows that AP2A2 is nearly ubiquitous, although it is excluded from one population. I suppose it is possible that the mRNA is only translated in a population of nociceptors, and these are the CGRP-positive nociceptors.

Although we had difficulty locating the Hockley et al. publication, we were able to locate the Sharma et al. publication the reviewer has referenced (PMID: 31915380). This reference is another outstanding piece of work that surveys mRNA expression changes in somatosensory neurons as they develop and mature in mice. The authors' use of scRNA-seq resulted in the generation of a large data repository available to the public (https://kleintools.hms.harvard.edu/tools/springViewer_1_6_dev.html?datasets/Sharma2019/all.) A quick glance at the database would suggest that there may be a baseline level of mRNA expression in nearly all sequenced cell clusters. However, we would still argue that there is a preferential expression of AP2 α 2 as it displays a level of clustering in distinct neuronal subpopulations in fact more so than Nav1.7. Clearly, Nav1.7 is not ubiquitously expressed in DRG neurons, as determined by numerous investigators, and this points to perhaps a limitation of RNAseq types of databases. We agree with the reviewer that looking solely at mRNA expression does not correlate to protein expression in all these cell types.

Nav1.7 (Scn9a) clustering

AP2α2 clustering